# Mediator kinase inhibition suppresses hyperactive interferon signaling in Down syndrome

Kira A Cozzolino[1], Lynn Sanford[2,3], Samuel Hunter[2,3], Kayla Molison[1], Benjamin Erickson[4,5], Meaghan CS Courvan[1,2,3,6,7], Taylor Jones[1], Deepa Ajit[8], Matthew D Galbraith[7,9], Joaquín M Espinosa[7,9], David Bentley[4,5], Mary Ann Allen[3], Robin D Dowell[2,3]*, Dylan J Taatjes[1]*

[1]Department of Biochemistry, University of Colorado, Boulder, United States; [2]Department of Molecular, Cellular, and Developmental Biology, University of Colorado, Boulder, United States; [3]BioFrontiers Institute, University of Colorado, Boulder, United States; [4]Department of Biochemistry and Molecular Genetics, University of Colorado School of Medicine, Aurora, United States; [5]UC-Denver RNA Bioscience Initiative, Aurora, United States; [6]Crnic Institute Boulder Branch, Boulder, United States; [7]Linda Crnic Institute for Down Syndrome, University of Colorado Anschutz Medical Campus, Aurora, United States; [8]Metabolon Inc, Durham, Morrisville, United States; [9]Department of Pharmacology, University of Colorado Anschutz Medical Campus, Aurora, United States

*For correspondence:
robin.dowell@colorado.edu
(RDD);
taatjes@colorado.edu (DJT)

## eLife Assessment

This is an **important** study providing **compelling** evidence that the Mediator kinase module mediates an elevated inflammatory response, manifested by heightened cytokine levels, associated with Downs syndrome (DS) via transcriptional changes impacting cell signaling and metabolism that involve mobilization of nuclear receptors by altered lipid metabolites, which has significance for the treatment of DS and other chronic inflammatory conditions. Particular strengths of the study include the combined experimental approaches of transcriptomics, untargeted metabolomics and cytokine screens and the use of sibling matched cell lines (trisomy 21 vs disomy 21) from various donors. Evidence is also provided implicating the Mediator kinase module in controlling mRNA splicing and mitochondrial function that should stimulate new research to elucidate the mechanistic bases for these novel functions.

**Abstract** Hyperactive interferon (IFN) signaling is a hallmark of Down syndrome (DS), a condition caused by Trisomy 21 (T21); strategies that normalize IFN signaling could benefit this population. Mediator-associated kinases CDK8 and CDK19 drive inflammatory responses through incompletely understood mechanisms. Using sibling-matched cell lines with/without T21, we investigated Mediator kinase function in the context of hyperactive IFN in DS over a 75 min to 24 hr timeframe. Activation of IFN-response genes was suppressed in cells treated with the CDK8/CDK19 inhibitor cortistatin A (CA), via rapid suppression of IFN-responsive transcription factor (TF) activity. We also discovered that CDK8/CDK19 affect splicing, a novel means by which Mediator kinases control gene expression. To further probe Mediator kinase function, we completed cytokine screens and metabolomics experiments. Cytokines are master regulators of inflammatory responses; by screening 105 different cytokine proteins, we show that Mediator kinases help drive IFN-dependent cytokine responses at least in part through transcriptional regulation of cytokine genes and receptors.

Metabolomics revealed that Mediator kinase inhibition altered core metabolic pathways in cell type-specific ways, and broad upregulation of anti-inflammatory lipid mediators occurred specifically in kinase-inhibited cells during hyperactive IFNγ signaling. A subset of these lipids (e.g. oleamide, desmosterol) serve as ligands for nuclear receptors PPAR and LXR, and activation of these receptors occurred specifically during hyperactive IFN signaling in CA-treated cells, revealing mechanistic links between Mediator kinases, lipid metabolism, and nuclear receptor function. Collectively, our results establish CDK8/CDK19 as context-specific metabolic regulators, and reveal that these kinases control gene expression not only via TFs, but also through metabolic changes and splicing. Moreover, we establish that Mediator kinase inhibition antagonizes IFN signaling through transcriptional, metabolic, and cytokine responses, with implications for DS and other chronic inflammatory conditions.

## Introduction

Down syndrome (DS) is relatively common in the human population (ca. 1 in 700 live births) (*de Graaf et al., 2017*) and results from trisomy of chromosome 21 (T21). T21 manifests in myriad ways, including an increased propensity for autoimmune and neurological disorders, as well as elevated incidence of leukemia (*Hasle et al., 2000*; *Antonarakis, 2017*; *Antonarakis et al., 2020*). Notably, T21 also results in chronic immune dysregulation associated with hyperactivation of interferon (IFN) signaling (*Sullivan et al., 2016*; *Araya et al., 2019*; *Waugh et al., 2019*).

The chronic, hyperactive IFN response in DS can be attributed, at least in part, to the fact that four IFN receptors are encoded on chromosome 21: IFNAR1 and IFNAR2 for Type I IFNs (e.g. IFNβ, IFNγ), IFNGR2 for IFNγ, and the Type III subunit IL10RB for IFN $\lambda$ , which also serves as a subunit of the IL10 receptor. Numerous pathologies associated with DS, including autoimmune-related disorders, are considered direct consequences of chronic IFN pathway activation (*Madan et al., 2006*; *Mårild et al., 2013*; *Araya et al., 2019*; *Waugh et al., 2019*; *Bush et al., 2021*). For these reasons, hyperactive IFN signaling lies at the heart of DS pathophysiology (*Waugh et al., 2023*), and therapeutic strategies to dampen IFN responses are being tested in clinical trials (NCT04246372, NCT05662228).

We recently demonstrated that the Mediator kinases (CDK8 and its paralog CDK19) are drivers of inflammatory responses to the universal cytokine IFNγ (*Steinparzer et al., 2019*). This discovery has implications for DS, because hyperactive IFN signaling underlies many DS symptoms and because Mediator kinases are promising targets for molecular therapeutics, in part due to low toxicity of a selective inhibitor in mouse models (*Pelish et al., 2015*). Many studies have linked Mediator kinase activity to immune system function and inflammatory responses (*Bancerek et al., 2013*; *Chen et al., 2017*; *Johannessen et al., 2017*; *Yamamoto et al., 2017*; *Guo et al., 2019*; *Steinparzer et al., 2019*; *Hofmann et al., 2020*; *Freitas et al., 2022*), and CDK8/CDK19 inhibition can suppress autoimmune disease in animal models (*Akamatsu et al., 2019*). Mediator kinases target the STAT transcription factor (TF) family (*Bancerek et al., 2013*; *Poss et al., 2016*) and activation of JAK/STAT pathway TFs (e.g. STAT1, IRF1) is blocked upon Mediator kinase inhibition in IFNγ-stimulated human or mouse cells (*Steinparzer et al., 2019*). Collectively, these results suggest that Mediator kinase inhibition could mitigate chronic, hyperactive IFN signaling in T21.

Mediator is a 26-subunit complex that regulates RNA polymerase II (RNAPII) transcription genome-wide (*Richter et al., 2022*). Mediator is recruited to specific genomic loci through interactions with sequence-specific, DNA-binding TFs, and Mediator interacts with the RNAPII enzyme at transcription start sites. Through these distinct interactions, Mediator enables TF-dependent control of RNAPII function. CDK8 or CDK19 can reversibly associate with Mediator as part of a 4-subunit 'Mediator kinase module' that contains additional subunits MED12, MED13, and CCNC (*Luyties and Taatjes, 2022*).

Mediator kinases have not been studied in the context of DS, and a goal of this project was to define their roles in the context of IFNγ signaling, using donor-derived cell lines. We also sought to address fundamental questions regarding Mediator kinase function that remained largely unexplored. For instance, it is not known whether Mediator kinases impact pre-mRNA splicing that is coupled to RNAPII transcription, despite evidence that CDK8 and/or CDK19 phosphorylate splicing regulatory proteins (*Poss et al., 2016*). Furthermore, metabolic changes are at least as important as transcriptional changes in driving biological responses (*Rinschen et al., 2019*; *Zhu and Thompson, 2019*),

but how Mediator kinases impact metabolism in human cells remains incompletely understood (*Zhao et al., 2012*; *Galbraith et al., 2017*). Finally, cytokines serve as key drivers of IFNγ responses, and despite links between CDK8/CDK19 function and inflammation (*Chen et al., 2017*; *Steinparzer et al., 2019*; *Freitas et al., 2022*), it is largely unknown whether Mediator kinases impact cytokine levels.

Using a combination of approaches, we have identified novel and diverse mechanisms by which Mediator kinases control IFNγ responses under both basal and IFNγ-stimulated conditions. Beyond effects on the RNAPII transcriptome, our results indicate that Mediator kinases act by governing TF activation and suggest that they selectively control gene expression through pre-mRNA splicing; furthermore, regulation of cytokine and metabolite levels contributes to CDK8/CDK19-dependent control of inflammatory signaling, including downstream activation of nuclear receptors through context-specific, kinase-dependent changes in lipid metabolites. Taken together, our findings identify Mediator kinases as therapeutic targets that could mitigate immune system dysregulation in individuals with DS.

## Results

### Experimental overview: transcriptomics, metabolomics, and cytokine screens

The goals of this project were to compare and contrast transcriptional and metabolic changes in the following experimental contexts: (1) T21 vs. D21, (2) ±IFNγ stimulation, (3) ±Mediator kinase inhibition, and (4) IFNγ stimulation + Mediator kinase inhibition. For T21 vs. D21 comparisons, we selected lymphoblastoid cells from the Nexus Biobank that were derived from siblings matched in age (3- or 5-year old) and sex (male). We used the natural product cortistatin A (CA) to inhibit CDK8 and CDK19 (*Pelish et al., 2015*). CA is the most potent and selective Mediator kinase inhibitor available (*Clopper and Taatjes, 2022*); for example, kinome-wide screens showed no off-target kinase inhibition even at 1 μM, a concentration 5000 times greater than its measured $K_D$ of 0.2 nM (*Pelish et al., 2015*). Throughout this project, we used CA at a concentration of 100 nM, to minimize potential off-target effects. Immunoblots confirmed CA-dependent inhibition of STAT1 S727 phosphorylation, a known CDK8 substrate (*Figure 1—figure supplement 1A*).

Prior to completion of RNA-seq experiments, we probed the timing of the IFNγ response at IFN target genes GBP1 and IRF1 by RT-qPCR (*Figure 1—figure supplement 1B*). Based upon these results, we chose the 4-hr time point for RNA-seq (see Methods). We also completed experiments in T21 cells after 24 hr ±CA. An overview of the RNA-seq experiments is shown in *Figure 1A*, with data for all genes across all conditions in *Figure 1—source data 1*.

Large-scale, untargeted metabolomic analyses were completed 24 hr post-IFNγ stimulation in D21 and T21 cells. This time point was chosen to approximate a 'steady state' following IFNγ treatment and to allow time for metabolic adaptations to occur; this time point was also consistent with prior metabolite analyses in IFN-treated cells (*Wang et al., 2018*; *Lee et al., 2021b*). An overview of the metabolomics experiments is shown in *Figure 1A*, with raw data for all identified metabolites across conditions in *Figure 1—source data 2*.

DS has been described as a 'cytokinopathy' based upon evaluation of blood plasma from individuals with or without T21. Arrays screening 29 different cytokines showed elevated levels in DS individuals (*Malle et al., 2023*). Based upon these and other results, we sought to directly measure cytokine levels to assess IFNγ- and CA-dependent effects. We completed a series of cytokine screens (*n* = 105 different cytokines) in T21 and D21 cells (*Figure 1—figure supplement 1C*), using the same experimental design as the metabolomics experiments (basal conditions, +CA, +IFNγ, or +CA+IFNγ). Biological replicate cytokine screens were completed for each condition after 24 hr treatment; measurements for all cytokines in both replicates can be found in *Figure 1—source data 3*.

### The T21 transcriptome, metabolome, and cytokine levels are consistent with hyperactive IFN signaling

For RNA-seq, biological triplicate experiments were completed, and a normalization (*Hunter et al., 2023*) was performed to account for the potential 1.5-fold expression differences from chromosome 21 in T21 cells (*Hwang et al., 2021*). Comparison of gene expression in T21 vs. D21 cells revealed massive differences, as expected (*Figure 1—figure supplement 1D*; *Figure 1—source data 1*). Gene

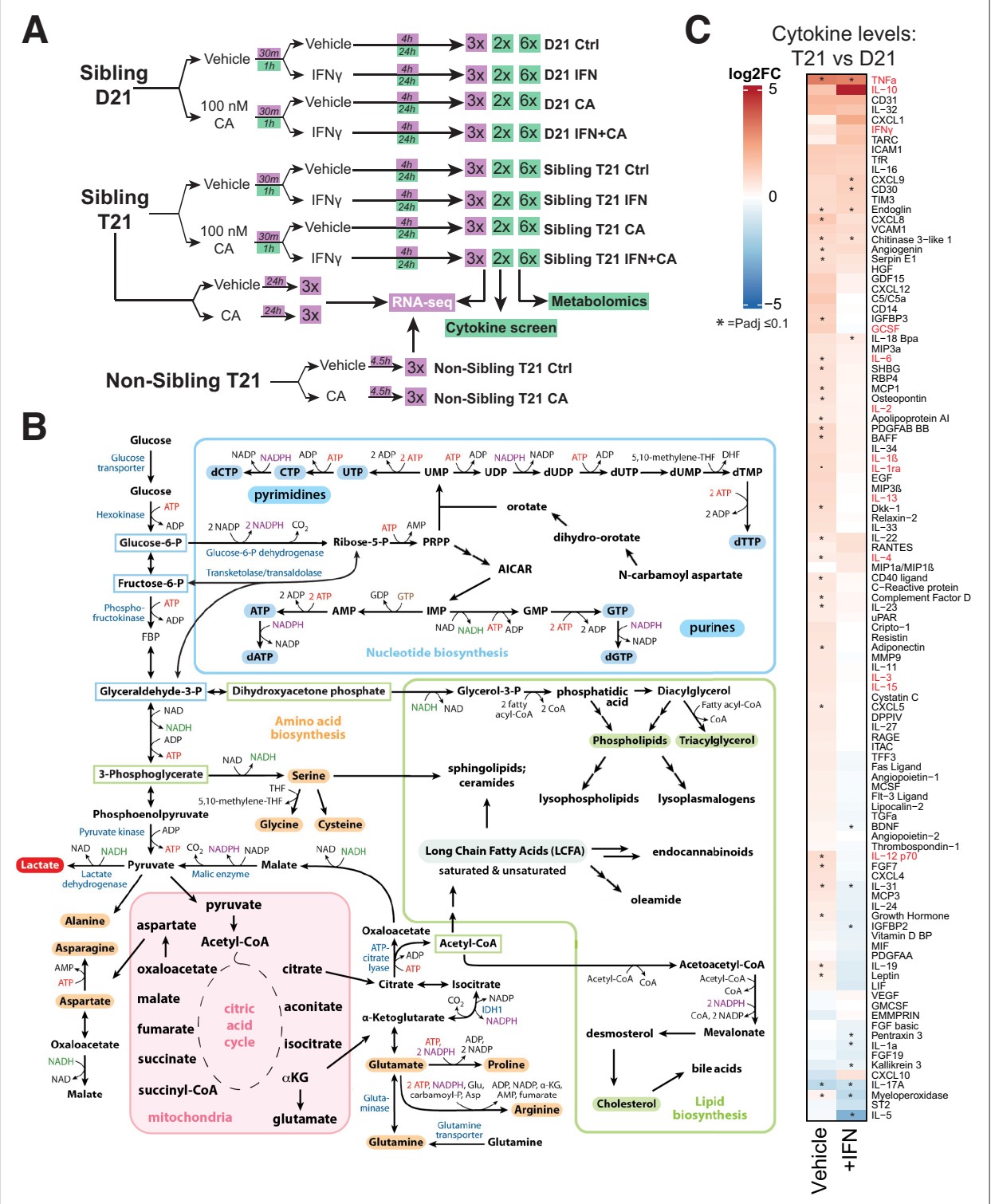

**Figure 1.** Experimental overview; elevated cytokines in T21 cells. (**A**) Schematic of cell treatment and data collection workflow for metabolomics, cytokine screen (green shading) and RNA-seq (purple shading). (**B**) Simplified diagram of human metabolic pathways, with an emphasis on those relevant to this study. Figure adapted from *Lunt and Vander Heiden, 2011*. (**C**) Heatmap of all cytokines measured (*n* = 105), comparing relative levels in vehicle-treated T21 cells vs. vehicle-treated D21 cells (left column) and relative levels in IFNγ-treated T21 cells vs. IFNγ-treated D21 cells (right column). Asterisk denotes padj ≤ 0.1.

The online version of this article includes the following source data and figure supplement(s) for figure 1:

*Figure 1 continued on next page*

*Figure 1 continued*

**Figure supplement 1.** Comparisons between T21 and D21 cells.

**Source data 1.** RNA-seq data summary.

**Source data 2.** Metabolomics data.

**Source data 3.** Cytokine screen data.

**Source data 4.** Gene set enrichment analysis (GSEA) results.

**Source data 5.** Ingenuity Pathway Analysis (IPA) results.

**Source data 6.** GO pathway analysis involving comparisons with Human Trisome Project large cohort study.

**Figure supplement 2.** Metabolic differences in T21 vs. D21; data comparison with whole blood cohort clinical studies.

set enrichment analysis (GSEA)(*Subramanian et al., 2005*) revealed significant upregulation of inflammatory pathways (e.g. IFNγ, IFNα, Complement, TNFα) in T21 vs. D21 cells (*Figure 1—figure supplement 1E*; *Figure 1—source data 4*), consistent with prior reports (*Sullivan et al., 2016*; *Sullivan et al., 2017*; *Malle et al., 2023*). Furthermore, the 'upstream regulators' identified from Ingenuity Pathway Analysis (IPA) (*Krämer et al., 2014*) predicted activation of numerous inflammatory markers in T21 cells (e.g. IFNγ, IFNα, TNF, TGFβ1, NFκB), based upon the RNA-seq results (*Figure 1—figure supplement 1F*; *Figure 1—source data 5*). Negatively enriched pathways in T21 cells (GSEA Hallmarks, *Figure 1—figure supplement 1E*, *Figure 1—source data 4*) reflected proliferative gene expression programs (e.g. MYC targets, G2M checkpoint); in agreement, we observed slower growth rates for T21 cells compared with their D21 counterparts.

Untargeted metabolomics experiments identified and quantified over 600 biochemicals representing every major metabolic pathway (*Figure 1—source data 2*). We observed evidence that inflammatory pathways were activated in T21 cells under basal conditions (IPA results, *Figure 1—figure supplement 1G*; *Figure 1—source data 5*), and 196 biochemicals increased and 214 decreased in T21 vs. D21 (*Figure 1—figure supplement 2A, B*; *Figure 1—source data 2*). Pathways related to fatty acid transport, growth, and energy metabolism were reduced in T21 cells (vs. D21; *Figure 1—figure supplement 1G*). Supporting these general trends, the metabolomics data revealed T21-specific changes in glycolysis, nucleotide and lipid biosynthesis, and other fundamental cellular processes (*Figure 1—figure supplement 2B*; *Figure 1—source data 2*). Metabolic distinctions between T21 and D21 cells will be described in more detail later, in the context of IFNγ treatment and/or Mediator kinase inhibition. An overview of metabolic pathways that are relevant for this study is shown in *Figure 1B*.

As with the RNA-seq and metabolomics data, the cytokine screen results implicated chronic activation of the IFN response in T21 cells under basal conditions; for example, levels of many cytokines were elevated in T21 vs. D21 cells (*Figure 1C*, left column). Furthermore, treatment with IFNγ appeared to 'normalize' cytokine levels in T21 vs. D21 cells (*Figure 1C*, right column), such that many cytokines were detected at roughly equal levels in IFNγ-stimulated T21 and D21 cells.

## T21-specific transcriptional, metabolic, and cytokine changes are reflected in population-level data

For practical reasons, the transcriptomics, metabolomics, and cytokine screening experiments completed here precluded parallel examination of additional genetically distinct donor-derived T21 and D21 cell lines (but see below). A basic observation from past studies was that the transcriptomes of T21 individuals are not only tissue- and cell type-specific, but also reflect individual-to-individual differences within the same cell types (*Sullivan et al., 2016*; *Hwang et al., 2021*). However, we hypothesized that the general trends in our sibling-matched T21 vs. D21 lines would show some commonality with large-scale cohort studies.

We analyzed RNA-seq data from whole blood samples from individuals with DS (*n* = 304) or euploid (i.e. D21; *n* = 96) controls (*Waugh et al., 2023*), generated by the Human Trisome Project (https://www.trisome.org/, NCT02864108). The data revealed upregulation of the IFNγ response, Complement, cytokine production, and other inflammatory pathways in T21 individuals (vs. D21; *Figure 1—figure supplement 2C*; *Figure 1—source data 4*). Approximately 10% of differentially expressed genes were shared between the Human Trisome Project cohort study (T21 vs. D21; whole

blood transcriptomes from individuals varying in age and sex) and our RNA-seq results from sibling-matched T21 and D21 lymphoblastoid cell lines (*Figure 1—figure supplement 2D*), consistent with individual-to-individual variation within the human population (*Sullivan et al., 2016*; *Hwang et al., 2021*). However, the shared 'core sets' of genes reflected prominent trends in our T21/D21 RNA-seq comparisons. For example, upregulated genes represented cytokine production and other inflammatory pathways, whereas downregulated genes involved nucleotide and fatty acid metabolism, splicing, G-protein-coupled receptor (GPCR) signaling, and proliferation. These pathway trends are well-represented in our RNA-seq and metabolomics data (e.g. *Figure 1—figure supplement 1E–G*; *Figure 1—source data 6*) and additional data supporting these T21-specific changes (vs. D21) will be described in more detail later.

We also used the IFN score metric (*Galbraith et al., 2023*) to evaluate our sibling-matched T21 and D21 lines. The IFN score considers expression of 18 genes from a set of 382 genes comprising the IFNγ, IFNα, and inflammatory response GSEA Hallmarks gene sets. It has been used to evaluate differences between whole blood transcriptomes in a cohort of 502 participants ($n$ = 356 T21, $n$ = 146 D21). As shown in *Figure 1—figure supplement 2E*, the sibling-matched T21 line had an elevated IFN score, consistent with comparative trends from a large cohort of whole blood transcriptomes of T21 or D21 individuals (*Galbraith et al., 2023*).

A recent analysis of blood plasma cytokine levels, measuring 29 cytokines from DS individuals ($n$ = 21) or age-matched euploid controls ($n$ = 10) identified elevated levels of approximately 10 to as many as 22 of these 29 cytokines (*Malle et al., 2023*). In agreement, at least 13 of the 22 cytokines (59%) were elevated in our T21 vs. D21 comparison (red font in *Figure 1C*). Collectively, the RNA-seq and cytokine data from a large cohort of DS individuals indicate that the two model cell lines evaluated here, derived from age- and sibling-matched D21 and T21 individuals, broadly reflect population-wide trends.

## Mediator kinase inhibition tempers T21 inflammatory pathways under basal conditions

We next asked whether Mediator kinases influenced hyperactive IFNγ signaling in T21 vs. D21 cells. We first describe how Mediator kinase inhibition impacted T21 or D21 cells under basal conditions.

### Transcriptome

As shown in *Figure 2A*, *Figure 2—figure supplement 1A–D*, and *Figure 1—source data 4*, CA caused similar changes to gene expression programs in T21 and D21 cells, although some T21-specific differences were evident (*Figure 2—figure supplement 1C, D*). CA treatment upregulated 445 genes in D21 (331 in T21; 210 shared) and downregulated 233 genes (191 in T21, 112 shared; padj 0.01). GSEA showed downregulation of MYC targets, consistent with prior studies in CDK8-depleted cells (*Sooraj et al., 2022*). Downregulation of the inflammatory 'TNFα signaling via NFκB' pathway was also observed in CA-treated D21 and T21 cells (*Figure 2—figure supplement 1C, D*); decreased expression of the NFKB1 and NFKB2 transcripts themselves likely contributed to this effect (*Figure 1—source data 1*). Regulation of NFκB transcriptional programs has been previously linked to Mediator kinases (*Chen et al., 2017*) although direct control of NFκB transcription levels has not been reported. Activation of GSEA Hallmarks related to mTOR signaling, cholesterol and fatty acid metabolism were observed in CA-treated D21 and T21 cells (*Figure 2—figure supplement 1C, D*), in agreement with prior studies in other cell types (*Poss et al., 2016*; *Audetat et al., 2017*; *Andrysik et al., 2021*). The genes significantly upregulated in these pathways are highlighted in *Figure 2—figure supplement 1E*. Importantly, the gene expression changes matched the metabolome changes in CA-treated cells (see below).

We next completed an 'upstream regulators' analysis (IPA; *Figure 2—figure supplement 1F*, *Figure 1—source data 5*) based upon the RNA-seq data. In agreement with the GSEA results, factors regulating cholesterol homeostasis (e.g. SCAP, SREBF1/2, INSIG1; up), fatty acid metabolism (e.g. ACSS2, FASN; up), and inflammation (e.g. NFKB1, MYD88, IL1B; down) were significantly altered by CA treatment in T21 cells (*Figure 2B*; *Figure 2—figure supplement 1F*), with similar results in D21 (*Figure 2—figure supplement 1F*). The IPA results also implicated CA-dependent repression of pro-inflammatory TFs specifically in T21 cells, including RELA (NFκB complex), FOXO3, and ILF3

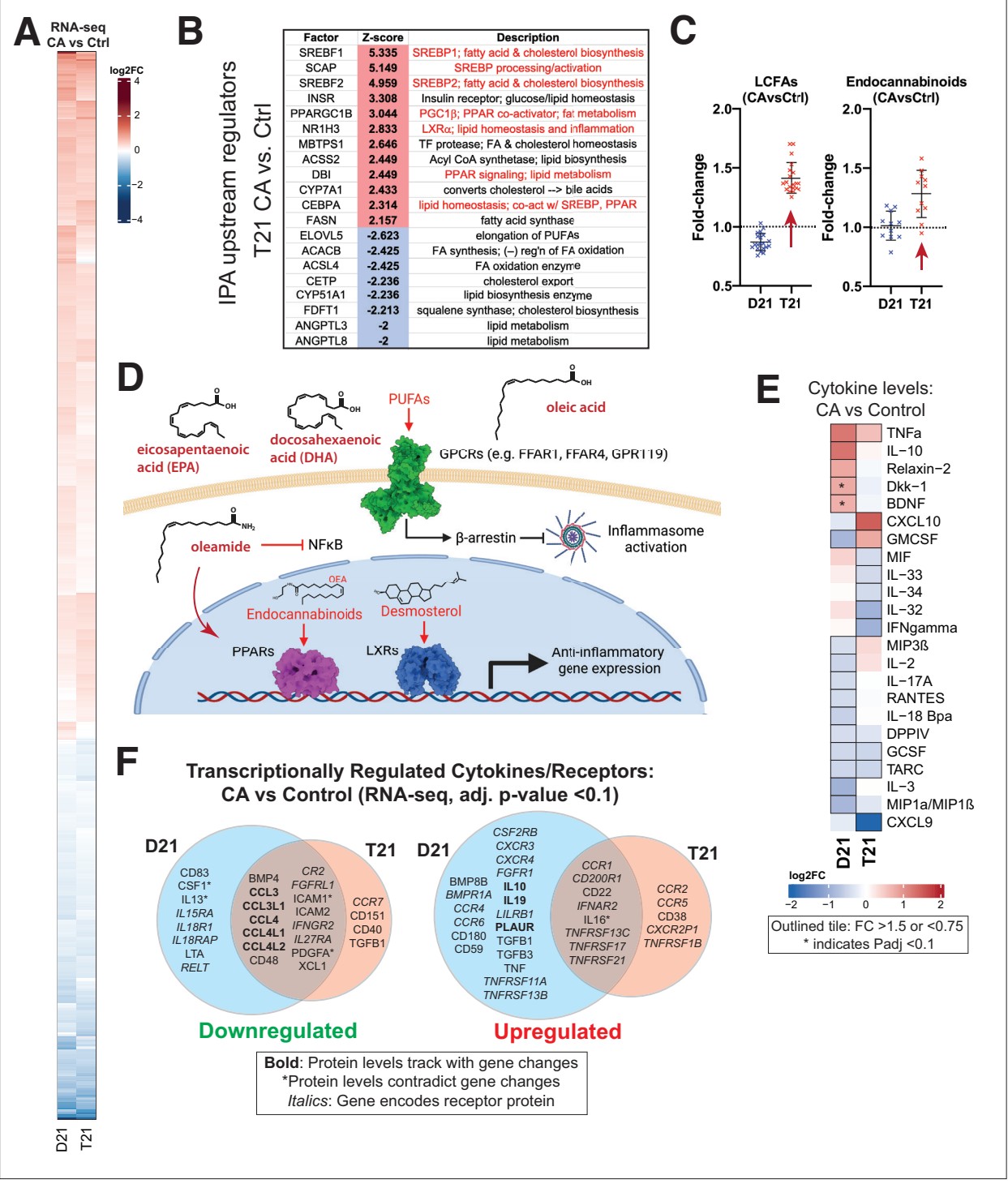

**Figure 2.** Mediator kinase inhibition tempers inflammatory pathways under basal conditions, with T21-specific effects. (**A**) Heatmap of genes with differential expression in D21 or T21 cells treated with cortistatin A (CA) (100 nM) compared to DMSO controls. Genes with adjusted p-value <0.01 in one or both cell lines are shown. (**B**) Table of activation Z-scores for selected upstream regulators predicted for gene expression changes in CA-treated T21 cells relative to DMSO controls. (**C**) Box plots showing relative levels of selected lipid metabolites in CA-treated D21 and T21 cells compared to DMSO controls. (**D**) Simplified diagram of pathways through which selected lipid metabolites regulate inflammation. (**E**) Heatmap of average relative cytokine levels in CA-treated cells compared to DMSO controls. Only cytokines with relative levels of ≥1.5 (log2FC; red shading) or ≤0.75 (blue shading) in one or both cell lines are shown; cytokines meeting this threshold are outlined in black. Asterisk (*) denotes adjusted p-value <0.1 using ANOVA. (**F**) Venn diagrams of cytokines and cytokine receptor genes (in *italics*) that were downregulated (FC <0.8, left diagram) or upregulated (FC >1.2, right

*Figure 2 continued on next page*

*Figure 2 continued*

diagram) in CA-treated D21 and T21 cells, compared to DMSO controls. Cytokines with matching trends from RNA-seq (4 hr) are listed in bold, whereas cytokines with inverse trends in at least one cell type are marked with an asterisk.

The online version of this article includes the following source data and figure supplement(s) for figure 2:

**Source data 1.** Metabolomics data sibling-matched T21 vs. D21.

**Source data 2.** Metabolomics data sibling-matched T21 and D21 ±CA.

**Figure supplement 1.** Cortistatin A (CA) suppresses T21 inflammatory pathways under basal conditions.

**Figure supplement 2.** Persistent activation of lipid metabolism and associated transcription factors (TFs) in cortistatin A (CA)-treated T21 cells; reduced expression of RNA processing factors.

**Figure supplement 3.** Cortistatin A (CA) treatment suppresses oxygen consumption rate (OCR) and extracellular acidification rate (ECAR) in T21 cells.

**Figure supplement 4.** Elevated oxygen consumption rate (OCR) and extracellular acidification rate (ECAR) in T21 vs. D21; cortistatin A (CA) treatment normalizes mitochondrial function and ECAR toward D21 levels.

**Figure supplement 5.** Cortistatin A (CA) treatment increases expression of lipid biosynthesis genes and suppresses inflammation and IFN-related genes in a different T21 donor line.

---

(*Figure 2—figure supplement 1F*; *Figure 1—source data 5*), consistent with repression of chronic basal IFN signaling upon Mediator kinase inhibition.

## Metabolome

CA-dependent changes in the metabolomes of D21 and T21 cells were strikingly distinct. This is likely due to (1) the longer timeframe used for metabolomics analyses (24 vs. 4.5 hr for RNA-seq), and (2) the massive 'pre-existing' basal metabolome differences in T21 vs. D21 cells (*Figure 2—source data 1*). Here, we will focus on the T21 vs. D21 metabolic differences that relate to pathways that were impacted by Mediator kinase inhibition.

In *Figure 2C*, we highlight examples of CA-dependent increases in anti-inflammatory lipid metabolites. In T21 cells selectively, CA treatment broadly increased levels of long chain mono- and poly-unsaturated FA (PUFA; *Figure 2C*), including oleic acid, eicosapentaenoate (EPA), docosapentaenoate (DPA), docosahexaenoate (DHA), and various other $\omega 3$ and $\omega 6$ PUFAs (*Figure 2—source data 2*). These metabolites act as signaling molecules by binding to extracellular GPCRs to trigger anti-inflammatory signaling cascades (*Basil and Levy, 2016*; *Husted et al., 2017*; *Kimura et al., 2020*; *Jordan and Werz, 2022*), as shown in the simplified schematic in *Figure 2D*. Similar to LCFAs, endocannabinoids were broadly elevated in CA-treated T21 cells (vs. D21; *Figure 2C*). Endocannabinoids mediate anti-inflammatory effects in part through PPAR TFs (*Alhouayek and Muccioli, 2014*) for example, oleoylethanolamide (OEA) and oleamide are PPARα ligands (*Fu et al., 2003*; *Roy et al., 2016*) and OEA can suppress pro-inflammatory NFκB signaling (*Yang et al., 2016*).

Consistent with lipid-dependent activation of LXR and PPAR nuclear receptors, the RNA-seq results showed evidence of CA-dependent LXR and PPAR activation specifically in T21 cells (*Figure 2B*; *Figure 2—figure supplement 1F*), which correlated with the elevated levels of LCFAs, desmosterol, oleamide, and endocannabinoids in CA-treated T21 cells (*Figure 2C*). Although metabolites were measured 24 hr after CA treatment, these data suggest that altered lipid metabolites influence LXR and PPAR function within 4.5 hr; moreover, evidence for PPAR activation in CA-treated T21 cells was also observed from RNA-seq data 24 hr after CA treatment (see below). Collectively, these results implicate lipid metabolites as key 'downstream' regulators of the transcriptional response to Mediator kinase inhibition. Importantly, this link between lipid metabolites and LXR or PPAR activation was observed in T21 but not D21 cells, suggesting that this CA-dependent effect selectively occurs in the context of hyperactive IFN signaling.

## Cytokine

CA treatment was also shown to be anti-inflammatory based upon the cytokine screen results. Under basal conditions, the levels of about two dozen cytokines changed in response to CA treatment (*Figure 2E*). In D21 cells, the anti-inflammatory cytokine IL10 increased upon Mediator kinase inhibition, in agreement with prior reports in other cell types (*Johannessen et al., 2017*), whereas the pro-inflammatory cytokines GMCSF, IL3, and MIP1α/β (a.k.a. CCL3/4) decreased; an exception was

TNFα. Similar results were obtained in T21 cells; however, CA treatment reduced the levels of pro-inflammatory cytokines in T21 cells that were not reduced in D21 (e.g. IL32, IL33, IL34), although CXCL10 and GMCSF were exceptions. The selective CA-dependent decrease in IFNγ in T21 cells under basal conditions (*Figure 2E*) further supports the chronic high baseline IFN signaling in these cells (*Sullivan et al., 2016*). Note that some cytokines cannot be clearly defined as either pro- or anti-inflammatory and their roles may be cell type-specific (*Dinarello, 2007*).

Because cytokine protein levels were measured 24 hr after CA treatment, we hypothesized that their altered levels would partially correspond to gene expression changes that were measured earlier, at 4.5 hr post-treatment. Indeed, this was observed for many cytokines impacted by Mediator kinase inhibition, as shown in *Figure 2F*. (Note that expression changes for cytokine receptors are also shown.)

## T21 transcriptome shows elevation of lipid biosynthesis genes and reduced levels of splicing regulators with 24 hr CA treatment

We next assessed longer-term effects of Mediator kinase inhibition on the transcriptome in T21 cells. We completed biological triplicate RNA-seq experiments in T21 cells 24 hr after CA treatment (or DMSO controls; *Figure 2—figure supplement 2A*). As shown in *Figure 2—figure supplement 2B*, GSEA Hallmarks showed little evidence for suppression of anti-inflammatory pathways, in contrast to the results in T21 cells after 4.5 hr CA treatment. This was further supported by GSEA with the GO Biological Processes (GOBP) pathway set (*Figure 2—figure supplement 2C*), which includes more gene sets ($n$ = 7641 for GOBP vs. $n$ = 50 for Hallmarks; see Methods) and more with relevance to IFN signaling and inflammation.

Gene sets related to cholesterol and lipid metabolism remained activated in CA-treated cells after 24 hr, consistent with the RNA-seq data at the shorter 4.5 hr timeframe. Moreover, the $t$ = 24 hr RNA-seq results showed evidence for sustained activation of SREBP as well as PPAR and LXR in CA-treated cells, a result that was inferred from the metablomics data (*Figure 2D*). For instance, IPA upstream regulators showed activation of 'PPARA:RXRA coactivator' and NR1H3/LXRA (*Figure 2—figure supplement 2D*). This coincided with elevated levels of PPAR and LXR ligands (e.g. OEA, oleamide, desmosterol), as measured by metabolomics 24 hr after CA treatment (*Figure 2C, D*; *Figure 2—source data 2*). We also generated GOBP heatmaps comparing the 4.5 vs. 24 hr ±CA RNA-seq data (*Figure 2—figure supplement 2E*). The results are distinct from the dot plots shown in *Figure 2—figure supplement 2C* because the heatmaps are not narrowly focused on selected GOBP terms and because a cutoff (padj ≤ 0.1) was applied. The gene sets change over time, underscoring the importance of timing in evaluation of kinase inhibitors. Also noteworthy were the many pathways related to RNA processing and splicing in the CA-treated cells at 24 hr. Splicing defects in CA-treated cells are described in a later section, but the results summarized in *Figure 2—figure supplement 2E* suggest that CA-dependent splicing changes are caused by reduced expression of splicing regulators, at least at longer timeframes.

## Mediator kinases influence T21 mitochondrial function

GSEA results from RNA-seq data in T21 cells after 24 hr CA treatment (*Figure 2—figure supplement 2B*) showed downregulation of oxidative phosphorylation, a result consistent with a prior proteomics study completed in HCT116 cells ±CA (*Poss et al., 2016*). To probe further, we measured the oxygen consumption rate (OCR) in D21 or T21 cells ±CA ($t$ = 24 hr). As shown in *Figure 2—figure supplement 3A, B*, OCR was reduced in CA-treated D21 or T21 cells, and reached statistical confidence in T21 cells. The reduced OCR in CA-treated T21 cells tracked with reduced mRNA levels for oxidative phosphorylation genes, measured by RNA-seq (*Figure 2—figure supplement 3C*). We also measured extracellular acidification rate (ECAR) in D21 or T21 cells ±CA ($t$ = 24 hr). Extracellular acidification is elevated by lactate, an end-product of glycolysis. As shown in *Figure 2—figure supplement 3D, E*, CA treatment reduced ECAR in both D21 and T21 cells, a result consistent with our metabolomics data (lactate reduced in CA-treated cells; *Figure 2—source data 2*) and prior CDK8-dependent links to expression of glycolysis enzymes (*Galbraith et al., 2017*).

A comparison of mitochondrial function in sibling-matched D21 and T21 cells is shown in *Figure 2—figure supplement 4A, B* and ECAR is shown in *Figure 2—figure supplement 4C*. The data show significant differences in T21 cells; these results are consistent with a T21 mouse model study, although

we note that T21/D21 OCR differences may be tissue-specific (*Sarver et al., 2023*). Because CA treatment lowered OCR and ECAR, and reduced mitochondrial respiration in T21 cells (*Figure 2—figure supplement 3B*), we next compared D21 cells with CA-treated T21 cells. The results suggest that Mediator kinase inhibition shifts T21 mitochondrial function and ECAR toward D21 levels (*Figure 2—figure supplement 4D–F*); however, we emphasize that this represents only one pair of cell lines and more work is needed to rigorously assess this topic.

## Similar CA-dependent transcriptome changes in different donor T21 cell line

Although completing parallel metabolomic, cytokine, and transcriptomics analyses across many cell lines was impractical, we did complete RNA-seq experiments in a different T21 donor cell line. Biological triplicate RNA-seq experiments were performed in a non-sibling T21 line ±CA (*t* = 4.5 hr). Similar to the 'sibling-matched' T21 line, CA-treated cells showed elevated expression of genes in the cholesterol homeostasis, fatty acid metabolism, and peroxisome pathways (*Figure 2—figure supplement 5A, B*). Moreover, GSEA completed with the GOBP set showed suppression of pathways related to IFN signaling and inflammation (*Figure 2—figure supplement 5C, D*). In fact, the CA-dependent suppression was amplified in this 'non-sibling' T21 line, which may reflect higher basal IFN signaling compared with the sibling-matched T21 line. IPA upstream regulators results (*Figure 2—figure supplement 5E*) also showed evidence for CA-dependent suppression of IFN signaling, with negative Z-scores for pro-inflammatory TFs (e.g. STAT2, IRF1, NFKB1) and cytokines (e.g. IFNG, IFNA2, IFNL1). Conversely, SREBP TFs were activated, as were LXR and PPAR, similar to the sibling-matched T21 line (*Figure 2—figure supplement 5E*). A heatmap summarizing transcriptome changes upon CA treatment in the two different T21 donor cell lines is shown in *Figure 2—figure supplement 5F*. Taken together, the transcriptomics data from this 'non-sibling' T21 cell line showed robust CA-dependent suppression of inflammatory pathways, with evidence for CA-dependent activation of SREBP, PPAR, and LXR TFs, consistent with the results from the different sibling-matched T21 line.

## Mediator kinase inhibition blocks IFNγ-induced transcriptional, metabolic, and cytokine responses

We next evaluated the transcriptomic, metabolomic, and cytokine data from IFNγ-treated cells. As shown in *Figure 3—figure supplement 1A*, the transcriptional response to IFNγ was broadly similar in the sibling-matched T21 and D21 cells; however, T21 cells showed higher basal expression of inflammatory genes. This can be seen upon comparison of gene expression in T21 vs. D21 cells under basal (*Figure 3—figure supplement 1B*) or IFNγ-stimulated conditions (*Figure 3—figure supplement 1C–E*). Only 10 genes showed greater induction in T21 cells in response to IFNγ, whereas 68 genes had reduced induction in T21 compared with D21 (*Figure 3—figure supplement 1F*; *Figure 1—source data 1*). Pathway analysis (GOBP) of these 68 genes showed predominant roles in IFN signaling and inflammatory responses (*Figure 3—figure supplement 1G*; *Figure 1—source data 6*). This result is consistent with hyperactive IFN signaling in T21 cells (*Sullivan et al., 2016*), such that induction upon exogenous IFNγ stimulation was reduced compared with D21 cells. Next, we focus on how Mediator kinase inhibition influenced transcriptional, metabolic, and cytokine responses in IFNγ-treated D21 and T21 cells.

### Transcriptome

The regulatory roles of the Mediator kinases CDK8 and CDK19 are context- and cell type-specific, and their functions appear to be especially important for initiating changes in gene expression programs, such as during an acute stimulus (*Luyties and Taatjes, 2022*). As shown in *Figure 3A*, many of the mRNAs responsive to IFNγ (activated or repressed) were impacted in the opposite way in CA + IFNγ-treated D21 or T21 cells (see also *Figure 3—figure supplement 2A, B*). This CA-dependent antagonism of IFNγ-response genes is best illustrated with difference heatmaps that compared levels of IFN-dependent transcriptional changes with IFN + CA conditions (i.e. IFN vs. Ctrl – IFN + CA vs. Ctrl; *Figure 3—figure supplement 2C, D*). An example at the CXCL9 gene locus is shown in *Figure 3B*; CXCL9 is a pro-inflammatory cytokine. GSEA also reflected CA-dependent suppression of the transcriptional response to IFNγ. In both D21 and T21 cell lines, IFNγ response and other inflammatory pathways were downregulated with CA treatment compared with IFNγ alone (*Figure 3C, D*;

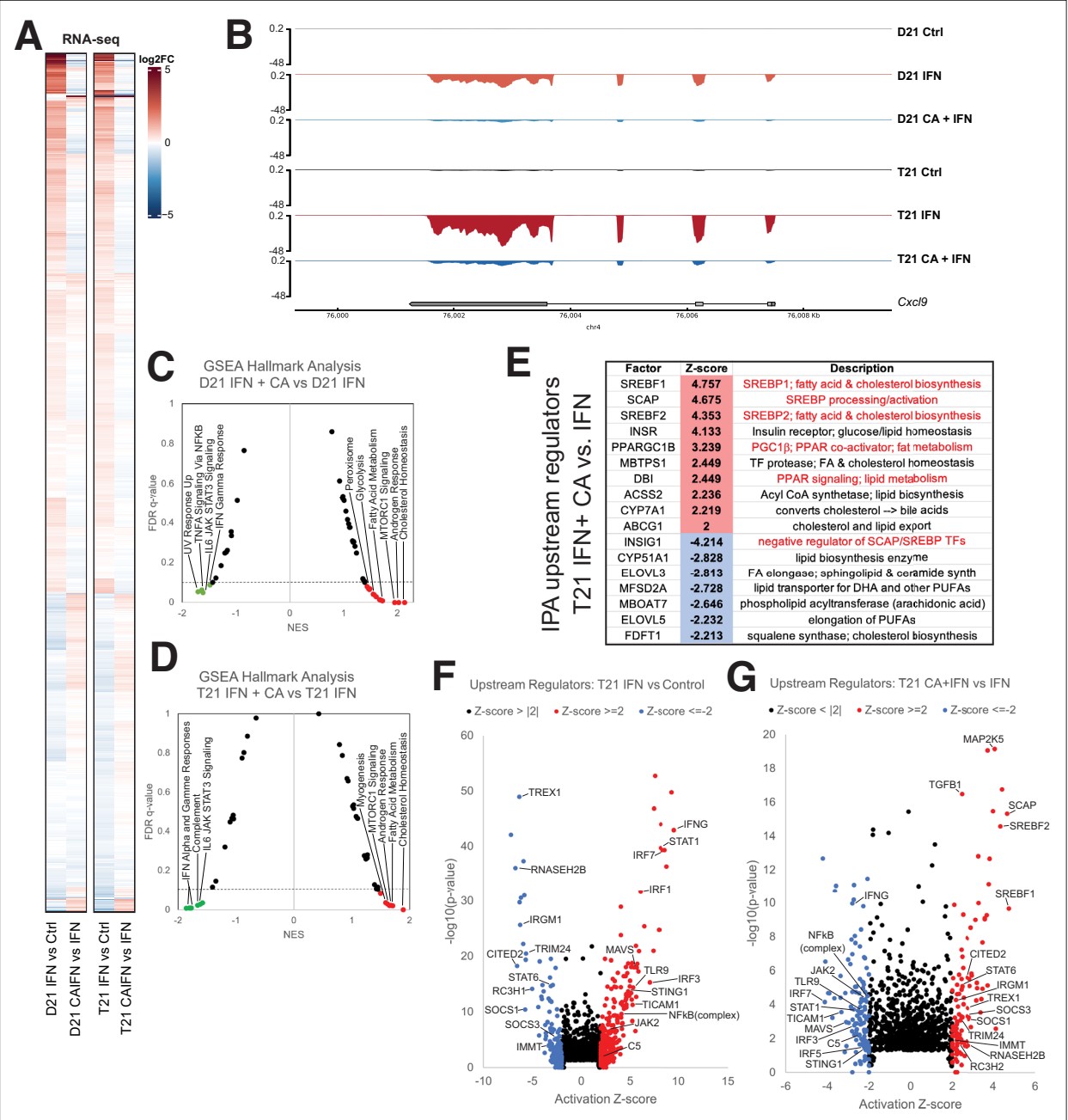

**Figure 3.** Mediator kinase inhibition antagonizes IFNγ transcriptional responses in T21 and D21. (**A**) Heatmap comparing gene expression patterns (RNA-seq) in IFNγ-treated D21 or T21 cells ±CA. This comparison shows cortistatin A (CA)-specific effects during IFNγ stimulation, which broadly counter changes caused by IFNγ alone. Genes with statistically significant (padj < 0.01) levels in one or both cell lines in IFNγ vs. control comparisons are shown. (**B**) Representative genome browser tracks for CXCL9 locus in D21 and T21 cells treated with vehicle, IFNγ, or IFNγ + CA. Gene set enrichment analysis (GSEA) moustache plots of Hallmark pathways in D21 (**C**) and T21 (**D**) cells treated with IFNγ + CA compared to IFNγ alone. This shows CA-specific effects during IFNγ stimulation. (**E**) Table of activation Z-scores for selected upstream regulators in CA-treated T21 cells treated with IFNγ + CA relative to IFNγ alone. This curated set emphasizes lipid metabolite changes. Ingenuity Pathway Analysis upstream regulators results from comparison of differential gene expression in T21 cells during IFNγ treatment (**F**) or in cells treated with IFNγ + CA compared to IFNγ alone (**G**). Selected transcription factors (TFs) and other factors associated with inflammatory responses are labeled. Analysis used only genes with padj < 0.1.

The online version of this article includes the following figure supplement(s) for figure 3:

**Figure supplement 1.** The response to exogenous IFNγ is suppressed in T21 vs. D21 cells.

**Figure supplement 2.** Mediator kinase inhibition antagonizes IFNγ transcriptional responses.

*Figure 3—figure supplement 2E*; *Figure 1—source data 4*). Similar results were obtained with an assessment of inflammatory pathways using IPA (*Figure 3—figure supplement 2F*; *Figure 1—source data 5*). Thus, Mediator kinase inhibition blocked normal transcriptional responses to IFNγ, consistent with prior studies in Murine Embryonic Fibroblasts (MEFs) and human cancer cells (*Bancerek et al., 2013*; *Steinparzer et al., 2019*). The upregulated Hallmark pathways included cholesterol homeostasis and fatty acid metabolism (*Figure 3C and D*; *Figure 1—source data 4*). A set of factors from the IPA upstream regulators analysis underscores this result, identifying TFs and enzymes involved in cholesterol and lipid metabolism as activated in CA-treated T21 cells (*Figure 3E–G*), with similar results in D21 (*Figure 3—figure supplement 2G, H*; *Figure 1—source data 5*). Furthermore, an array of pro-inflammatory TFs were induced in IFNγ-treated T21 and D21 cells, as expected; however, activation of these same TFs was blocked in CA-treated cells (D21 and T21; *Figure 3F, G*; *Figure 3—figure supplement 2G, H*). These results suggest that inhibition of CDK8/CDK19 suppresses IFNγ responses, in part, through inhibition of pro-inflammatory TF activity, in agreement with prior experiments (*Steinparzer et al., 2019*).

To probe further, we compared the GSEA 'leading edge' IFN-response gene set from a whole blood transcriptome dataset generated by the Human Trisome Project (*Waugh et al., 2023*) with the CA-responsive genes in our T21 cell line. As shown in *Figure 3—figure supplement 2I*, 86% of these genes (69 out of 80) overlapped with those identified as elevated in the Human Trisome Project dataset, reflecting a common transcriptional response to IFNγ. Among these IFN-responsive genes were many whose expression decreased upon CA treatment, including STAT1, IRF1, GBP4, MX2, XCL1, and CXCL9. Moreover, CA treatment decreased the IFN score (*Galbraith et al., 2023*) in IFN-stimulated T21 cells (*Figure 3—figure supplement 2J*). These results reveal that Mediator kinase inhibition suppresses transcriptional responses that are typically hyperactivated in DS individuals.

## Metabolome

Consistent with the RNA-seq results, metabolic changes in D21 and T21 cells reflected a CA-dependent suppression of IFNγ-induced inflammation. CA treatment reversed metabolic changes induced by IFNγ alone (*Figure 4A–C*). For example, LCFAs, endocannabinoids, oleamide, desmosterol, and bile acids were reduced in D21 cells under IFNγ stimulation conditions, consistent with the anti-inflammatory roles of these metabolites (*Alhouayek and Muccioli, 2014*; *Basil and Levy, 2016*; *Husted et al., 2017*; *Kimura et al., 2020*; *Jordan and Werz, 2022*). In contrast, their levels were elevated in CA-treated D21 cells. The metabolic effects of Mediator kinase inhibition were blunted in IFNγ-treated T21 cells, which may reflect the already elevated basal IFN signaling in T21 cells (*Figure 4A–C*; *Figure 4—source data 1*). In *Figure 4C*, we highlight the CA-dependent elevation of desmosterol and oleamide (LXR and PPAR ligands, respectively), and the CA-dependent decline in quinolinic acid, a tryptophan derivative that is typically elevated in DS (*Powers et al., 2019*). Collectively, the data summarized in *Figure 4A–C* are consistent with CA-dependent anti-inflammatory effects.

## Cytokines

Compared with basal conditions, many more cytokines showed CA-responsiveness in the context of IFNγ stimulation, and a majority of the changes countered IFNγ-induced inflammatory responses (*Figure 4—figure supplement 1A*). This CA-dependent antagonism of IFNγ cytokine responses is illustrated in the difference heatmaps shown in *Figure 4D, E*. Levels of many pro-inflammatory cytokines were reduced in IFNγ + CA-treated D21 and T21 cells, including CXCL9, C5, IL33, IL1α, and others (*Figure 4—figure supplement 1A*).

Because cytokine levels were measured 24 hr after treatment, we hypothesized that their altered levels would at least partially correspond to gene expression changes measured earlier, at 4.5 hr post-treatment. This was observed for many IFNγ-induced cytokines impacted by Mediator kinase inhibition, as shown in *Figure 4—figure supplement 1B*. (Note that expression changes for cytokine receptors are also shown.) Collectively, the cytokine screen demonstrated that inhibition of Mediator kinase function generally downregulated pro-inflammatory cytokines while upregulating anti-inflammatory cytokines (e.g. IL10, LIF, IL19). In the context of DS, this could potentially mitigate pathological immune system hyperactivation.

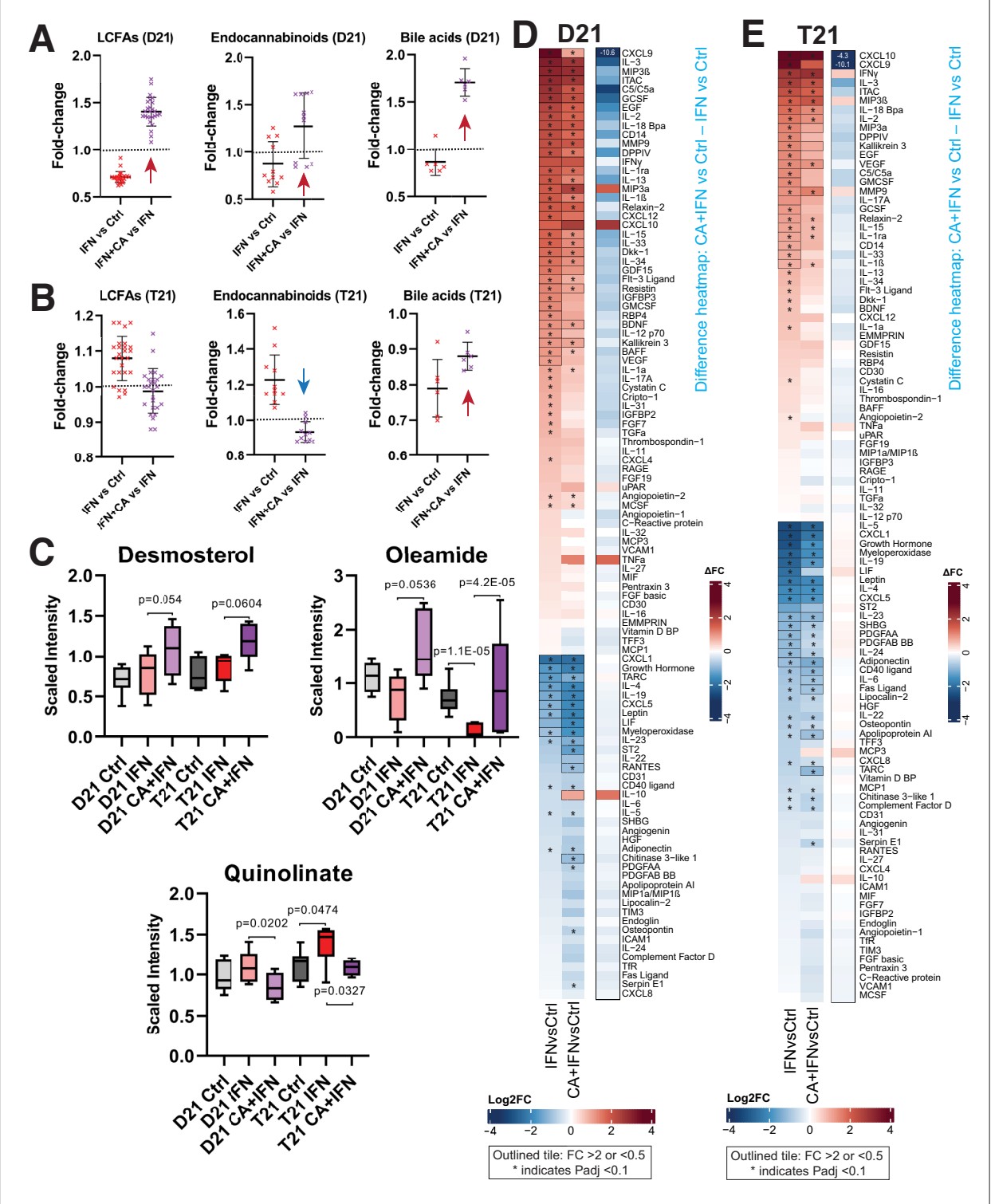

**Figure 4.** Mediator kinase inhibition reverses pro-inflammatory metabolic and cytokine changes triggered by IFNγ. Effect of IFNγ treatment on select classes of lipid metabolites is shown (IFNγ vs. Ctrl) alongside the effect of cortistatin A (CA) treatment in IFNγ-treated cells, in D21 (**A**) and T21 (**B**) cells. The LCFAs represented here include saturated, mono- and poly-unsaturated FA, shown in rows 3–28 in *Figure 4—source data 1*. Each point represents a different metabolite, with line and whiskers representing the mean and SD. Note CA treatment reverses IFNγ effects generally (arrows). (**C**) Box plots showing levels of anti-inflammatory metabolites desmosterol or oleamide, and pro-inflammatory metabolite quinolinate, in D21 or T21 cells treated with DMSO (Ctrl), IFNγ, or IFNγ + CA. Heatmaps showing changes in cytokine levels in D21 (**D**) and T21 (**E**) cells, after the indicated treatments. Only cytokines with relative levels of ≥2.0 (log2FC; red shading) or ≤0.5 (blue shading) in one or both cell lines are shown; cytokines meeting this threshold

*Figure 4 continued on next page*

*Figure 4 continued*

are outlined in black. Asterisk (*) denotes adjusted padj < 0.1 using ANOVA. Alongside each heatmap set (D21 or T21) is a 'difference' heatmap (IFN + CA vs. Ctrl – IFN vs. Ctrl levels) that highlights how Mediator kinase inhibition suppresses cytokine responses to IFNγ.

The online version of this article includes the following source data and figure supplement(s) for figure 4:

**Source data 1.** Metabolomics data sibling-matched T21 and D21 ±CA ±IFNγ.

**Figure supplement 1.** Mediator kinases transcriptionally enable cytokine responses to IFNγ.

## Mediator kinases regulate splicing in pathway- and cell type-specific ways

As Mediator-associated kinases, CDK8 and CDK19 are established regulators of RNAPII transcription (*Luyties and Taatjes, 2022*) however, their potential impact on splicing has not been examined. Over 95% of human mRNAs are alternatively spliced and defects in this process contribute to errors in gene expression in human diseases (*Scotti and Swanson, 2016*). We compared alternative splicing events using rMATS (*Shen et al., 2014*), separately evaluating all RNA-seq replicates (biological triplicates; *Figure 1A*). The data revealed 741 differential exon skipping events in T21 cells (vs. sibling-matched D21) under basal conditions (*Figure 5A, B*; *Figure 5—source data 1*), with approximately equal numbers of increased exon inclusion (*n* = 382) and increased exon skipping events (*n* = 359). A smaller number of other alternative splicing events (e.g. intron retention) were detected (*Figure 5—figure supplement 1A*). We observed similar trends in IFNγ-treated T21 and D21 cells, in which a greater number of exon skipping events (*n* = 418 vs. *n* = 296) occurred in IFNγ-stimulated T21 vs. D21 cells (*Figure 5—figure supplement 1B, C*; *Figure 5—source data 1*).

To determine whether Mediator kinase activity might influence splicing, we next evaluated CA-treated D21 and T21 cells. We found that CA treatment had a substantial impact on splicing, in both D21 and T21 cells (*Figure 5C*, *Figure 5—figure supplement 1D*). We identified 432 or 444 sites with altered exon skipping events (D21 or T21, respectively) in CA-treated cells compared with controls (ΔPSI [percent spliced in] ≥0.2, padj < 0.05; *Figure 5C*, *Figure 5—figure supplement 1E*; *Figure 5—source data 1*), and CA treatment increased inclusion of alternative exons more frequently than it increased their skipping in both lines (*n* = 239 vs. 193 in D21; *n* = 256 vs. 188 in T21). Similarly in IFNγ-treated D21 and T21 cells (*Figure 5D*, *Figure 5—figure supplement 1F*; *Figure 5—source data 1*), Mediator kinase inhibition favored exon inclusion (*n* = 178 in D21; 213 in T21) over exon skipping (*n* = 130 in D21; 148 in T21).

To assess whether the exon inclusion events sensitive to Mediator kinase function were selective for specific gene sets, we completed pathway analyses (IPA) across all conditions tested (*Figure 5E–G*, *Figure 5—figure supplement 1H, I*; *Figure 1—source data 5*). To increase statistical power, we grouped exon skipping events together (i.e. alternative exon inclusion events +alternative exon skipping events). Thus, the T21 vs. D21 comparison included 741 events, D21 +CA vs. Ctrl included 432 events, and so on. The results revealed that splicing changes triggered by Mediator kinase inhibition were cell type-specific (T21 vs. D21) and selectively impacted inflammatory and metabolic pathway genes. As shown in *Figure 5E*, genes associated with inflammatory signaling (red highlighted pathways) showed evidence of differential splicing in CA-treated T21 cells (vs. D21) during IFNγ stimulation. In addition, pyrimidine salvage and biosynthesis genes were alternatively spliced in CA-treated T21 cells (vs. DMSO; *Figure 5G*). These pathway-specific effects in CA-treated cells were consistent with gene expression changes (e.g. GSEA Hallmark gene sets, *Figure 1—source data 4*) and metabolic changes observed in CA-treated cells. For instance, the pyrimidine biosynthesis intermediates *N*-carbamoylaspartate, dihydro-orotate, and orotate were markedly depleted (1.4- to 3-fold) in CA-treated T21 cells (vs. DMSO; *Figure 1—source data 2*). The data in *Figure 5G* implicate CA-dependent splicing changes as a contributing factor.

Finally, we completed parallel splicing analyses for (1) the different donor T21 cell line ±CA treatment for 4.5 hr, as well as for (2) the sibling-matched T21 line with 24 hr CA treatment (vs. DMSO controls). We identified 349 sites with altered exon skipping events in the 4.5 hr CA-treated non-sibling T21 cells, and 309 sites in the sibling-matched T21 line at 24 hr ±CA (*Figure 5—figure supplement 2A, B*; *Figure 5—source data 2*). We next completed a pathway analysis (IPA) of genes with differential exon skipping (i.e. alternative exon inclusion events + alternative exon skipping events) ± CA (*Figure 5—source data 3*). For the sibling-matched line with 24 hr CA treatment, IPA identified

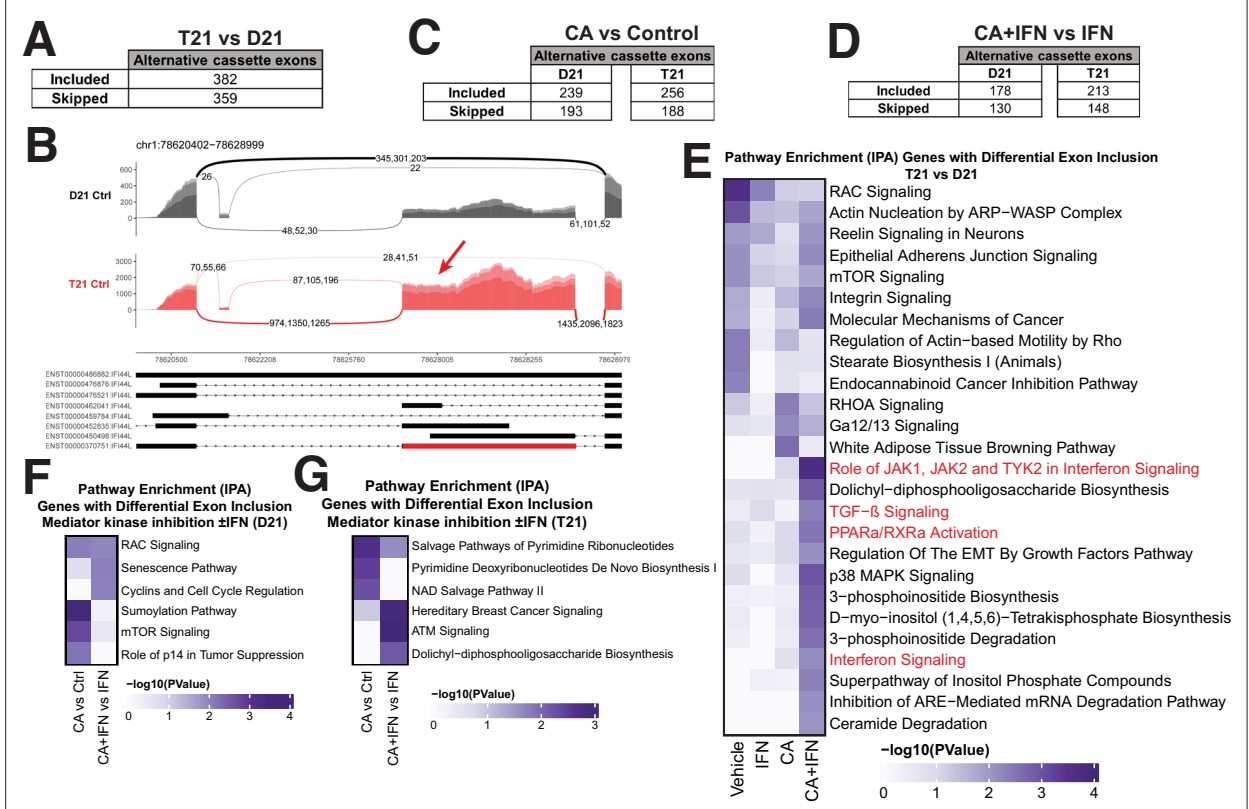

**Figure 5.** Mediator kinases regulate splicing in pathway-specific ways. (**A**) Table of alternative exon usage from untreated T21 cells compared to D21. Inclusion criteria were assessed at FDR <0.05, |InclusionLevelDifference| >0.2, and ≥2 reads/replicate. (**B**) Sashimi plots for the IFI44L gene, with normalized read numbers for D21 control (black) and T21 control (red) samples on the *y*-axis, and splice junction read numbers (representing each of three replicate experiments). (**C**) Table of alternative exon usage from cortistatin A (CA)-treated D21 or T21 cells compared to controls. Inclusion criteria were same as for panel A. (**D**) Table of alternative exon usage from D21 or T21 cells treated with IFNγ and CA compared to IFNγ alone. Inclusion criteria were same as for panel A. (**E**) Ingenuity Pathway Analysis enrichment results of genes with alternative exon skipping events in T21 vs. D21 cells; different treatment conditions indicated at bottom. Pathways relevant to IFNγ signaling are highlighted in red. Genes affected by alternative splicing in T21 vs. D21 cells could be grouped into different signaling pathways (e.g. RAC, mTOR, integrin) that are important for robust immune responses (*Jones and Pearce, 2017*; *Lee et al., 2021a*), suggesting how alternative splicing may influence inflammatory signaling in T21 cells. Ingenuity Pathway Analysis enrichment results of genes with alternative exon skipping events in D21 (**F**) or T21 (**G**) cells treated with CA (±IFNγ).

The online version of this article includes the following source data and figure supplement(s) for figure 5:

**Source data 1.** Splicing data (rMATS); sibling-matched T21 and D21.

**Source data 2.** Splicing data (rMATS); non-sibling T21 and sibling T21 *t* = 24 hr.

**Source data 3.** Ingenuity Pathway Analysis (IPA) results from splicing data.

**Figure supplement 1.** Additional information about splicing changes.

**Figure supplement 2.** Summary of splicing changes ±CA in non-sibling T21 (*t* = 4.5 hr) and sibling-matched T21 at 24 hr.

various metabolic and signaling cascades, including JAK–STAT, whereas the IPA results from the different donor T21 line (*t* = 4.5 hr ±CA) yielded results more similar to the other T21 line after 4.5 hr CA treatment (*Figure 5—figure supplement 2C, D*). In particular, IFN signaling pathways were represented, further implicating Mediator kinases as context-specific regulators of RNAPII transcription and splicing.

## Altered TF function underlies metabolic and anti-inflammatory effects in Mediator kinase-inhibited T21 cells

Sequence-specific, DNA-binding TFs are a major target of Mediator kinases, based upon quantitative phosphoproteomics data (*Poss et al., 2016*). To broadly assess how TF activity changed in response to Mediator kinase inhibition, we completed PRO-seq experiments in control vs. CA-treated T21 cells.

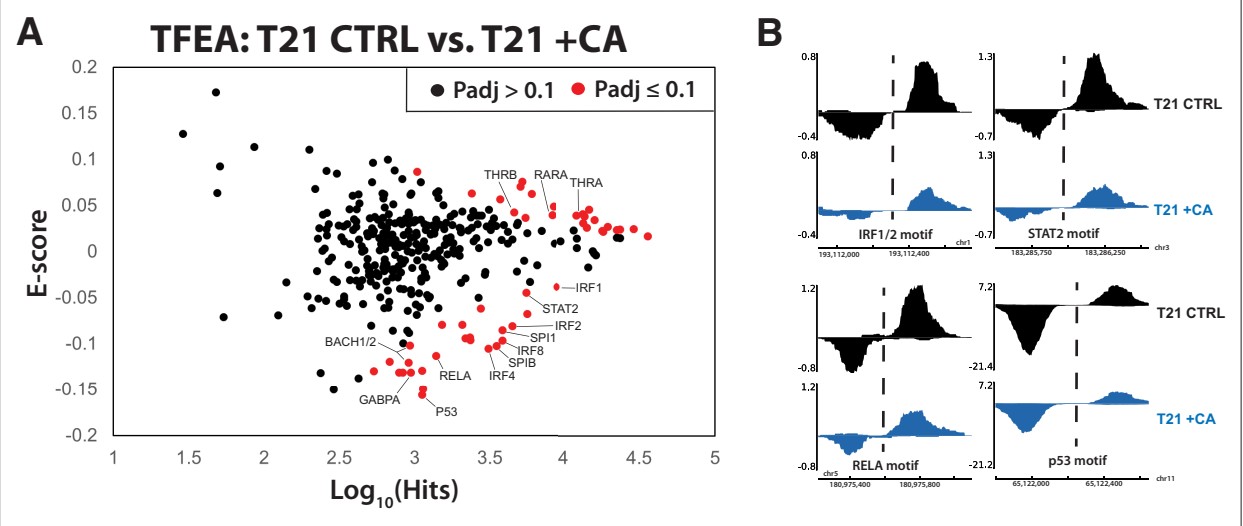

**Figure 6.** Mediator kinase inhibition rapidly suppresses pro-inflammatory transcription factors (TFs) in T21 cells. (**A**) MA plot of Transcription Factor Enrichment Analysis (TFEA) results comparing control (DMSO) vs. cortistatin A (CA)-treated T21 cells under basal conditions, from PRO-seq experiments completed after 75-min CA treatment. (**B**) Representative examples of bidirectional 'eRNA' transcription from PRO-seq data, showing reduced levels in CA-treated cells. Location of TF consensus binding motifs indicated with dashed line.

The online version of this article includes the following source data for figure 6:

**Source data 1.** Transcription Factor Enrichment Analysis (TFEA) results from PRO-seq data, T21 ±CA.

When sequenced at high depth (≥60 M reads/replicate), PRO-seq allows reliable detection of short, bidirectional transcripts that center around consensus TF-binding sites. Studies from numerous labs have shown that these bidirectional transcripts represent 'signatures' of active TFs bound to these sites (*Azofeifa et al., 2018*; *Wang et al., 2019*; *Kristjánsdóttir et al., 2020*; *Rubin et al., 2021*). Similarly, loss of bidirectional transcription indicates reduced TF activity.

Because our primary interest was CA-dependent suppression of hyperactive IFN signaling in DS, we completed PRO-seq experiments in T21 cells under basal conditions, ±CA treatment (*t* = 75 min). We then applied TFEA (Transcription Factor Enrichment Analysis), a well-tested method to globally assess TF activity from PRO-seq data (*Rubin et al., 2021*). As shown in *Figure 6A*, CA treatment broadly shut down pro-inflammatory TFs in T21 cells (e.g. IRF4, IRF8, SPIB, BACH1, RELA; examples shown in *Figure 6B*), consistent with the 'downstream' RNA-seq (4.5 hr), cytokine (24 hr), and metabolomics data (24 hr). Furthermore, it was evident that CA treatment rapidly repressed the p53 family of TFs (*Figure 6A, B*, *Figure 6—source data 1*). As a tumor suppressor, p53 represses cholesterol and lipid biosynthesis pathways, in part by blocking activation of SREBP (*Moon et al., 2019*). Rapid suppression of p53 by Mediator kinase inhibition is consistent with the metabolic changes observed in CA-treated T21 cells (*Figure 2C*). Moreover, the RNA-seq data implicated SREBP TF activation in CA-treated T21 cells vs. DMSO controls at later time points (4–24 hr), consistent with de-repression of SREBP following CA-dependent inhibition of p53 activity. From these results, we conclude that altered TF function is a major contributor to (1) CA-dependent metabolic changes and (2) suppression of pro-inflammatory IFN signaling in T21 cells.

## Discussion

Previous studies have shown that Mediator kinases regulate transcriptional responses to inflammatory stimuli (e.g. IFNγ or TNFα) in a variety of model systems (*Chen et al., 2017*; *Steinparzer et al., 2019*). Because individuals with DS have chronic, hyperactive IFN signaling (*Sullivan et al., 2016*), we hypothesized that Mediator kinase inhibition would antagonize inflammatory signaling cascades in T21 cells. This hypothesis was supported by the experimental results described here. To maintain focus on Mediator kinase activity, we avoided knockdown or knockout experiments, in part because CDK8/CDK19 proteins serve additional functions that are not kinase-dependent (*Poss et al., 2016*; *Audetat et al., 2017*; *Sooraj et al., 2022*; *Chen et al., 2023*). Collectively, our results expand upon

established concepts for CDK8 and CDK19 (e.g. regulation of IFNγ signaling) and reveal new insights about their impact on TF activation, metabolic and cytokine responses, as well as their effect on pre-mRNA splicing. Our results also establish Mediator kinases as potential therapeutic targets to suppress hyperactive IFN signaling in DS.

We acknowledge that individual-to-individual genetic variation could contribute to the differences we observe in our T21 vs. D21 comparisons (*Sullivan et al., 2016*; *Hwang et al., 2021*). We chose cell lines from sibling donors (both males of similar age) to minimize this possibility. Moreover, cross-referencing our results with data collected from large cohorts of individuals of varying age and sex (with/without T21) showed broadly similar trends in transcript (*Waugh et al., 2023*), metabolite (*Powers et al., 2019*), and cytokine levels (*Malle et al., 2023*) in the T21/D21 comparisons. Although our multi-omics T21/D21 comparisons yielded new insights, Mediator kinases were a focus of this study, and the T21/D21 comparisons primarily served as benchmarks to assess CDK8/CDK19 function in the context of DS and IFNγ signaling. A limitation of this study was that only two donor-derived T21 cell lines were evaluated, with one sibling-matched D21 line. Furthermore, the transcriptomic, metabolomic, and cytokine data were collected at select time points, such that some temporal trends in IFNγ responses can only be inferred (see below).

## CDK8 and CDK19 kinase activity impacts core metabolic pathways

Metabolomics has been described as the most direct readout of cell state, yielding a 'front line' assessment of the biochemicals that drive all cellular processes (*Rinschen et al., 2019*). The role of Mediator kinases in human cell metabolism is poorly understood. As a whole, our metabolomics data revealed the most striking CDK8/CDK19 effects on nucleotide biosynthesis (especially in D21) and lipid homeostasis. Nucleotide levels (purines and pyrimidines) and their intermediates were broadly reduced in CA-treated cells whereas different classes of lipids were generally elevated (*Figure 2— source data 2*).

The gene expression changes (RNA-seq, 4.5 or 24 hr) triggered by Mediator kinase inhibition tracked with these 'downstream' (24 hr) metabolic changes (e.g. *Figure 2B*, *Figure 2—figure supplement 1E*). For example, nucleotide biosynthesis mRNAs were downregulated whereas lipid metabolism mRNAs were upregulated in CA-treated cells; this was reflected in the GSEA Hallmarks, with positive enrichment for pathways such as cholesterol homeostasis, bile acid metabolism, and FA metabolism and negative enrichment for DNA repair pathways, which contains many nucleotide biosynthesis genes. The differential timing of the transcriptomics and metabolomics experiments helps link CA-dependent metabolic changes to gene expression changes; however, we cannot exclude other factors and pathways. For example, reduced levels of nucleotides can trigger cellular compensation through upregulation of FA levels (*Schoors et al., 2015*). Consequently, elevated levels of FA and other lipids may result, in part, from compensatory mechanisms triggered by CA-dependent reductions in nucleotide levels.

## Pathway-specific regulation of splicing by Mediator kinases

Despite widespread study of Mediator kinases as regulators of RNAPII transcription, their potential function in splicing has not been addressed. We hypothesized that CDK8 and/or CDK19 may impact splicing based upon prior results that identified NAB2, SRRM2, and KDM3A as high-confidence CDK8/CDK19 substrates in human cells (*Poss et al., 2016*) each of these proteins has been linked to regulation of splicing (*Gautam et al., 2015*; *Soucek et al., 2016*; *Baker et al., 2021*). Our analysis revealed that exon skipping was dependent upon the kinase function of CDK8 and/or CDK19; increased inclusion or skipping of hundreds of alternative exons was observed in CA-treated cells (D21 or T21), and this was observed under both basal and IFNγ-stimulated conditions (e.g. *Figure 5C, D*).

Pathway analysis of the CA-dependent alternative splicing events identified gene sets associated with signaling pathways known to be regulated by Mediator kinases, such as TGFβ and IFN signaling (*Figure 5E*; *Figure 5—figure supplement 2*). Interestingly, CA-dependent alternative splicing of IFN-related gene sets occurred specifically in T21 cells (not D21); moreover, other pathways were selectively affected by Mediator kinase inhibition in cell type and context-specific ways. For instance, mTOR pathway genes in CA-treated D21 cells and pyrimidine metabolism genes in T21 cells (*Figure 5F, G*).

Taken together, these results suggest that Mediator kinases regulate pre-mRNA splicing, which adds to the mechanisms by which CDK8 and CDK19 can influence gene expression. Our data further

suggest that kinase-dependent splicing regulation occurs in cell type- and context-specific ways, similar to other known regulatory functions for CDK8 and CDK19 (*Luyties and Taatjes, 2022*). We emphasize, however, that the effects of CDK8/CDK19 on splicing are modest, especially compared with CDK7, a different transcription-associated kinase (*Rimel et al., 2020*).

The mechanisms by which CDK8 and/or CDK19 influence pre-mRNA splicing remain unclear, but are likely to involve both direct kinase-dependent regulation of splicing factors (e.g. through phosphorylation of NAB2 or SRRM2) and indirect regulation through phosphorylation of DNA-binding TFs, which can regulate splicing (*Rambout et al., 2018*) and are common Mediator kinase substrates (*Poss et al., 2016*). Differential expression of splicing regulatory proteins in CA-treated cells may also contribute, based upon the reduced expression of splicing regulators at longer timeframes (*t* = 24 hr CA treatment; *Figure 2—figure supplement 2E*).

## Mediator kinase inhibition activates nuclear receptors through lipid metabolites

The anti-inflammatory lipids elevated in CA-treated cells included oleamide, desmosterol, endocannabinoids such as OEA, and PUFAs such as EPA, DPA, DHA, and various other $\omega 3$ or $\omega 6$ PUFAs. These metabolites act as signaling molecules, at least in part, through binding nuclear receptors or GPCRs such as FFAR1, FFAR4, and GPR119, to initiate a cascade of anti-inflammatory responses (*Fu et al., 2003*; *Basil and Levy, 2016*; *Husted et al., 2017*; *Kimura et al., 2020*; *Jordan and Werz, 2022*). Our RNA-seq results (4.5 and 24 hr) confirmed CA-dependent activation of nuclear receptors PPAR and LXR, but only under conditions with elevated IFN signaling (e.g. T21 cells or D21 +IFNγ). Furthermore, metabolomics data revealed elevated levels of PPAR and LXR ligands (e.g.

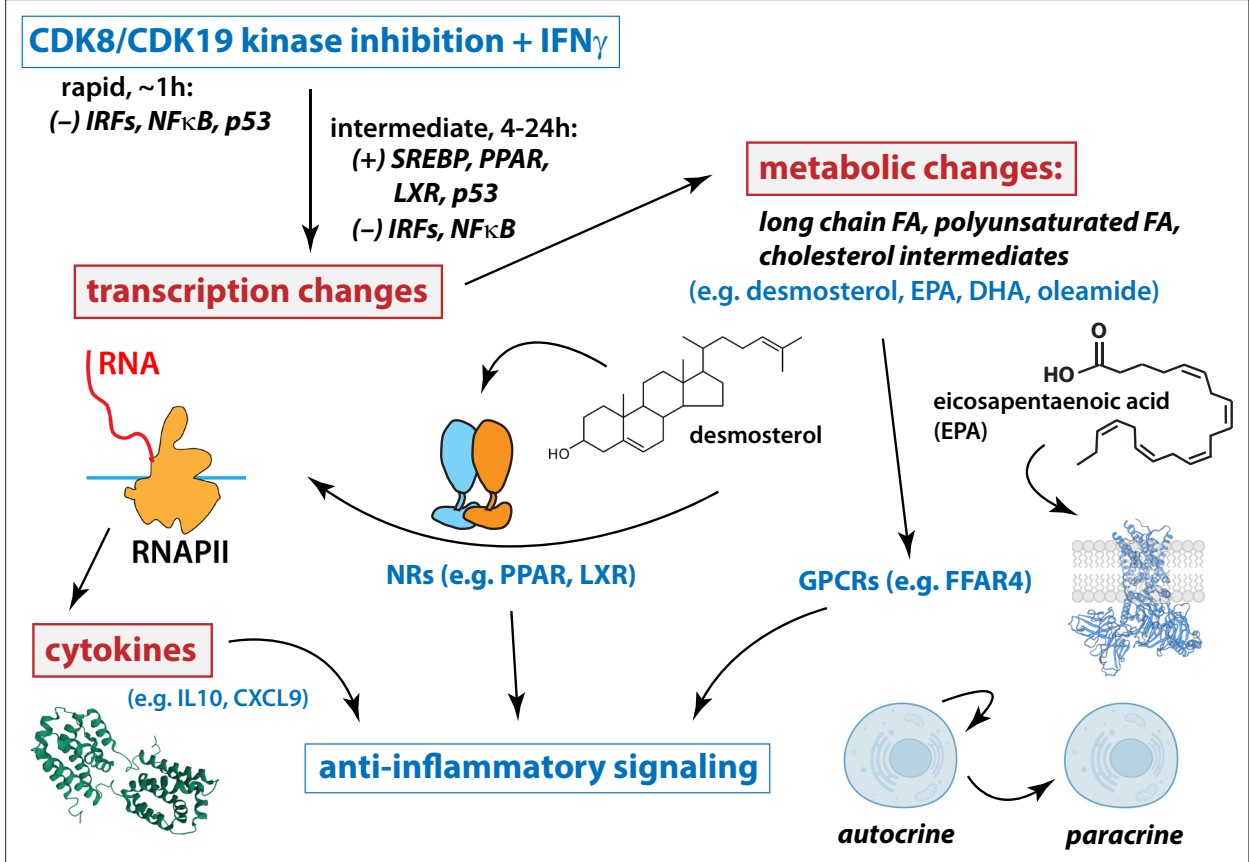

**Figure 7.** Working model for Mediator kinase-dependent regulation of IFNγ signaling. Under conditions of IFN stimulation, Mediator kinase inhibition suppresses pro-inflammatory transcriptional responses, including expression of cytokine genes and receptors. Mechanistically, this occurs through rapid inhibition of IRF and NF$\kappa$B transcription factors (TFs), which persists 4–24 hr after Mediator kinase inhibition. Mediator kinase inhibition also activates SREBP, which likely results from rapid suppression of p53 (*Moon et al., 2019*). Metabolic changes include increased levels of PPAR and LXR ligands, which tracks with activation of PPAR and LXR target genes 4–24 hr after Mediator kinase inhibition. See text for additional details.

oleamide, endocannabinoids, desmosterol) only in CA-treated cells in the context of hyperactive IFN signaling. These results reveal an unexpected mechanistic link between Mediator kinase activity and nuclear receptor function, and suggest potential regulation of GPCR function as well (*Figure 7*). Metabolic changes are generally not considered when evaluating the biological function of transcription-associated kinases such as CDK8 and CDK19. Our results demonstrate that kinase inhibition can activate TFs through metabolic changes, and suggest that during IFNγ stimulation, Mediator kinases normally suppress metabolic pathways that would activate PPAR and LXR nuclear receptors.

## Mediator kinase inhibition antagonizes IFN signaling

Independently and collectively, our transcriptomics, metabolomics, and cytokine screen data revealed that Mediator kinase inhibition suppressed pro-inflammatory signaling. This was most evident during IFNγ stimulation but was also apparent under basal conditions, especially in T21 cells due to their high 'baseline' activation of IFN signaling pathways. We acknowledge that metabolites and cytokines can have variable, dynamic, and context-specific effects on inflammatory signaling. For example, sphingolipids (*Maceyka and Spiegel, 2014*) and bile acids may serve pro- or anti-inflammatory roles depending upon cell type (*Chiang and Ferrell, 2019*), and cytokine responses are complex, in part because cytokines regulate each other (e.g. MIP3α/CCL20 is downregulated by IL10). For this study, a more straightforward metric for an anti-inflammatory change was whether CA treatment countered the effect of IFNγ stimulation alone. This was observed in a large number of cases across all experiments.

CDK8 and CDK19 are established regulators of RNAPII transcription (*Luyties and Taatjes, 2022*). Accordingly, the majority of CA-dependent effects that we have characterized here result from gene expression changes, consistent with our TFEA results (*Figure 7*) and prior phosphoproteomics data that identified TFs and other RNAPII regulatory factors as high-confidence targets of CDK8/CDK19 in human cells (*Poss et al., 2016*). The staggered timing of our PRO-seq (75 min), RNA-seq (4.5 or 24 hr), metabolomics (24 hr), and cytokine screen experiments (24 hr) allowed inference of direct vs. indirect effects of Mediator kinase inhibition. As expected, we observed changes in anti-inflammatory transcriptional responses over time, which likely reflects compensatory responses to Mediator kinase inhibition.

Taken as a whole, our results suggest that under conditions of IFN pathway stimulation, CDK8/CDK19 inhibition initiates a cascade of events that mediate anti-inflammatory responses (*Figure 7*). At short timeframes (75 min), Mediator kinase inhibition suppressed activation of IFN-responsive, pro-inflammatory TFs in T21 cells (*Figure 6*). Consistent with these results, mRNA levels of pro-inflammatory genes were subsequently reduced in CA-treated T21 cells (*Figure 2—figure supplement 5C, D*), as were protein levels of pro-inflammatory cytokines (*Figure 2E*). Similar CA-dependent effects were observed with exogenous IFNγ stimulation in both T21 and D21 cells (*Figures 3 and 4*). TFEA also identified p53 as rapidly suppressed in CA-treated T21 cells ($t$ = 75 min); p53 blocks SREBP activation and this contributes partially or completely to p53-dependent suppression of cholesterol and lipid biosynthesis (e.g. to suppress growth) (*Moon et al., 2019*). CA-dependent suppression of p53 function in T21 cells could therefore activate SREBP to elevate cholesterol and lipid biosynthesis. In agreement, RNA-seq data ($t$ = 4.5 or 24 hr) identified SREBP (IPA upstream regulators), cholesterol homeostasis and fatty acid metabolism (GSEA Hallmarks) as activated upon Mediator kinase inhibition. SREBP TFs activate cholesterol and lipid biosynthesis pathways, and metabolomics data revealed elevated levels of anti-inflammatory lipids in CA-treated cells during IFNγ signaling (e.g. D21 cells +IFNγ; T21 cells at basal conditions). Among these elevated lipids, oleamide, desmosterol, and PUFAs are nuclear receptor agonists that activate PPAR and LXR (*Yang et al., 2006*; *Sun and Bennett, 2007*; *Spann et al., 2012*; *Roy et al., 2016*); PPAR and LXR, in turn, broadly control fatty acid biosynthesis (*Schultz et al., 2000*), including PUFAs such as DPA, DHA, and EPA, whose levels were consistently elevated in CA-treated cells. In this way, Mediator kinase inhibition may robustly and durably elevate lipid metabolites, with potential downstream consequences for inflammatory signaling and nuclear receptor function (*Figure 7*).

## Concluding remarks

Although the factors that contribute to the DS condition are complex (*Antonarakis et al., 2020*), it has been described as both an interferonopathy (*Sullivan et al., 2016*) and a cytokinopathy (*Malle et al., 2023*). An emerging theme in DS research is that individuals with DS could benefit from therapeutic

strategies that diminish chronic, hyperactive IFN signaling (*Sullivan et al., 2017*; *Tuttle et al., 2020*; *Waugh et al., 2023*). Through multi-omics evaluation of CA-dependent effects in T21 cells, we have demonstrated that Mediator kinase inhibition antagonizes hyperactive IFN signaling, via diverse mechanisms that extend beyond transcription (*Figure 7*). Consequently, selective inhibition of Mediator kinase function represents a potential therapeutic strategy to address chronic inflammation and its associated co-morbidities in DS.

# Materials and methods
## Cell culture
Immortalized lymphoblastoid cell lines derived from age- and sex-matched sibling pairs (male; one D21, age 5.7 years: TIC0000259, one T21, age 3.1 years: TIC0000177) were obtained from the Nexus Biobank. The non-sibling T21 line was also obtained from the Nexus Biobank (male, age 10.7 years; TIC0001672). Cell lines were authenticated (STR profiling) by Nexus Biobank and tested negative for mycoplasma. Cells were cultured in RPMI medium (Gibco, 23400062) supplemented with 20% fetal bovine serum (Peak Serum Inc, PS-FB3), 1× GlutaminePlus (R&D Systems, R90210), and 1× penicillin–streptomycin (Gibco, 15140-163), on low-attachment flasks at 37°C and 5% $CO_2$. Due to the lower proliferation rate of T21 cells compared to D21 cells, T21 cells were seeded at twice the density of D21 cells for all experiments unless otherwise noted. For all CA treatments, the compound remained in the media during IFNγ stimulation.

## Western blotting
RIPA lysis buffer (Thermo Scientific, 89900) was used to generate cell lysates; all lysis was performed at 4°C in the presence of Halt Protease and Phosphatase Inhibitor Cocktail, EDTA-free (100×) (Thermo Scientific, 78441) and Benzonase endonuclease (Millipore Sigma, 101697). The BCA Protein Assay Kit (Pierce, 23227) was used to measure and normalize protein concentrations across samples. Protein samples were boiled for 5 min at 95°C in 4× Laemmli Sample Buffer (Bio-Rad, 1610747) and run on SDS-acrylamide gels, followed by transfer to nitrocellulose membranes. Blocking was performed using either 5% milk in TBS-T or in 5% bovine serum albumin for phospho-antibodies.

Antibodies used are as follows: anti-STAT1 (RRID:AB_2198300; Cell Signaling Technology, 9172), anti-phospho STAT1 S727 (RRID:AB_2197983; Cell Signaling Techology, 9177), and anti-GAPDH (RRID:AB_1080976; GeneTex, GTX100118). HRP-conjugated anti-rabbit secondary antibody was used for detection (Invitrogen, 31480) and protein was visualized using Immunobilon Western Chemiluminescent HRP Substrate (Millipore, WBKLS0500) on an ImageQuant LAS 4000 (GE Healthcare). Intensity values were quantified and normalized using ImageJ.

## RT-qPCR
Cells were seeded and allowed to recover for 24 hr prior to being treated with 10 ng/ml interferon gamma (Gibco PHC4031) or vehicle (40 mM Tris pH 7.4). Total RNA was isolated using TRizol (Invitrogen, 15596026) according to the manufacturer's instructions, quantified with a Qubit 3.0 using the RNA High Sensitivity (HS) kit (Invitrogen, Q32855), and 100 ng of RNA from each sample was converted to cDNA using High-Capacity cDNA Reverse Transcription Kit (Applied Biosystems, 4368813) following the manufacturer's instructions. cDNA was diluted to 0.25 ng/μl for all samples, and amplification was performed using Sybr Select Master Mix (Thermo Fisher Scientific, 4472908) on a Bio-Rad CFX384 Real Time PCR System. ΔΔCT values were calculated using GAPDH as a loading control. Treatments, RNA isolation, and RT-qPCR were performed in biological duplicate. The primers used for qPCR are as follows – GAPDH Forward: ACCACAGTCCATGCCATCAC, GAPDH Reverse: TCCACCACCCTGTTGCTGTA, GBP1 Forward: GTGCTAGAAGCCAGTGCTCGT, GBP1 Reverse: TGGGCCTGTCATGTGGATCTC, IRF1 Forward: GAGGAGGTGAAAGACCAGAGC, IRF1 Reverse: TAGCATCTCGGCTGGACTTCGA.

To probe the timing of the RNA-seq experiments, we completed RT-qPCR at various time points as shown in *Figure 1—figure supplement 1B*. The 4 hr time point (post-IFNγ) was chosen because robust induction of GBP1 and IRF1 mRNA was observed. Whereas mRNA levels remained high at 6 hr, we selected the 4 hr time point because it would better represent the primary transcriptional

response; moreover, the 4 hr time point was consistent with prior transcriptomics studies of IFNγ response in mammalian cells (*Steinparzer et al., 2019*).

## Metabolomics

For each treatment condition, six flasks of cells were seeded and allowed to recover for 24 hr prior to treatment. Cells were pre-treated with 0.1% DMSO or 100 nM CA for 1 hr, then treated with 40 mM Tris, pH 7.4 or 10 ng/ml interferon gamma (Gibco, PHC4031). After 24 hr of interferon gamma treatment, cells were harvested and snap-frozen in liquid nitrogen. Sample preparation was carried out by Metabolon (Durham, North Carolina, USA) using a previously published workflow (*Ford et al., 2020*). Samples were prepared using the automated MicroLab STAR system from Hamilton Company, and several recovery standards were added prior to the first step in the extraction process for QC purposes. To remove protein, dissociate small molecules bound to protein or trapped in the precipitated protein matrix, and to recover chemically diverse metabolites, proteins were precipitated with methanol under vigorous shaking for 2 min (Glen Mills GenoGrinder 2000) followed by centrifugation. The resulting extract was divided into five fractions: two for analysis by two separate reverse phase (RP)/UPLC–MS/MS methods with positive ion mode electrospray ionization (ESI), one for analysis by RP/UPLC–MS/MS with negative ion mode ESI, one for analysis by HILIC/UPLC–MS/MS with negative ion mode ESI, and one sample was reserved for backup. Samples were placed briefly on a TurboVap (Zymark) to remove the organic solvent.

All methods utilized a Waters ACQUITY ultra-performance liquid chromatography (UPLC) and a Thermo Scientific Q-Exactive high resolution/accurate mass spectrometer interfaced with a heated electrospray ionization (HESI-II) source and Orbitrap mass analyzer operated at 35,000 mass resolution. The sample extract was dried then reconstituted in solvents compatible to each of the four methods. Each reconstitution solvent contained a series of standards at fixed concentrations to ensure injection and chromatographic consistency. One aliquot was analyzed using acidic positive ion conditions, chromatographically optimized for more hydrophilic compounds. In this method, the extract was gradient eluted from a C18 column (Waters UPLC BEH C18-2.1 × 100 mm, 1.7 µm) using water and methanol, containing 0.05% perfluoropentanoic acid (PFPA) and 0.1% formic acid (FA). Another aliquot was also analyzed using acidic positive ion conditions, however it was chromatographically optimized for more hydrophobic compounds. In this method, the extract was gradient eluted from the same afore mentioned C18 column using methanol, acetonitrile, water, 0.05% PFPA, and 0.01% FA and was operated at an overall higher organic content. Another aliquot was analyzed using basic negative ion optimized conditions using a separate dedicated C18 column. The basic extracts were gradient eluted from the column using methanol and water, however with 6.5 mM Ammonium Bicarbonate at pH 8. The fourth aliquot was analyzed via negative ionization following elution from a HILIC column (Waters UPLC BEH Amide 2.1 × 150 mm, 1.7 µm) using a gradient consisting of water and acetonitrile with 10 mM Ammonium Formate, pH 10.8. The MS analysis alternated between MS and data-dependent MSn scans using dynamic exclusion. The scan range varied slighted between methods but covered 70–1000 *m/z*.

Identification of metabolites was performed through automated comparison of the ion features in experimental samples to a reference library of chemical standard entries (*Dehaven et al., 2010*), and are based on three criteria: retention index within a narrow RI window of the proposed identification, accurate mass match to the library ±10 ppm, and the MS/MS forward and reverse scores between the experimental data and authentic standards. The MS/MS scores are based on a comparison of the ions present in the experimental spectrum to the ions present in the library spectrum. While there may be similarities between these molecules based on one of these factors, the use of all three data points can be utilized to distinguish and differentiate biochemicals.

## Statistical and pathway analysis of metabolomics data

Two types of statistical analyses were performed: (1) significance tests and (2) classification analysis. Standard statistical analyses were performed in Array Studio on log-transformed data. For analyses not standard in Array Studio, the R program (http://cran.r-project.org/) was used. Following log transformation and imputation of missing values, if any, with the minimum observed value for each compound, Welch two-sample *t* test was used as a significance test to identify biochemicals that differed significantly (p < 0.05) between experimental groups. An estimate of the false discovery

rate (*q*-value) was calculated to take into account the multiple comparisons that normally occur in metabolomic-based studies. Classification analyses included principal component analysis, hierarchical clustering, and random forest.

Pathway analysis of metabolomic data was performed using the Ingenuity Pathway Analysis software (QIAGEN) (*Krämer et al., 2014*). For metabolomic analysis, a total of 566 out of 675 metabolites could be mapped using one of the following identifiers: CAS registry number, Human Metabolome Database (HMDB) identification, KEGG, PubChem CID, or RefSeq number; metabolites not used in IPA analysis can be found in *Figure 1—source data 2*. Metabolomic pathway analysis used only mapped metabolites with a p-value of <0.1 when making pathway predictions.

## Seahorse metabolic assays

Cells for metabolic assays were seeded 24 hr prior to treatments and allowed to recover, then were treated with either vehicle or CA at a final concentration of 100 nM. After 24 hr of treatment, cells were collected, washed, and resuspended in Seahorse XF RPMI Assay Media (Agilent, 103576) supplemented with 10 mM glucose (Agilent, 103577-100), 1 mM pyruvate (Agilent, 103578-100), and 2 mM glutamine (Agilent, 103579-100). Cells were counted and for each cell line 100,000 cells were seeded per well on XF24 cell culture plates that had been pre-coated with poly-D-lysine (Gibco, A3890401). The plates were spun down at $300 \times g$ for 3 min using the minimum brake setting, and cells were then allowed to recover for 30–60 min in a humidified, non-$CO_2$ incubator set to 37°C. Compounds from the Mito Stress Test Kit (Agilent, 103015-100) were used at the following final concentrations: 1.5 µM Oligomycin, 1 µM FCCP, and 5 µM Rotenone/Antimycin A. All assays were performed on an Agilent Seahorse XFe24 Analyzer (XFe version 2.6.3.5), and data were processed using the Agilent Wave software (version 2.6.3). Two to three technical replicates of each treatment condition were plated per assay, and three biological replicates of each assay were completed on different days. Across all three biological replicates, measurements from 2 wells that did not exhibit a drop in OCR of at least 30% following Oligomycin injection were considered outliers and were removed from analysis. Technical replicates from each assay were averaged, and the average OCR or ECAR values for each treatment condition or genotype at each time point were normalized relative to the first baseline measurement taken of vehicle-treated cells (for treatment comparisons) or D21 cells (for the cross-genotype comparisons). Relative values for OCR or ECAR at each time point, for each treatment condition, were averaged between all three biological replicates.

## RNA-seq

Cells were pre-treated with 0.1% DMSO or 100 nM CA for 30 min, then treated with 40 mM Tris, pH 7.4 or 10 ng/ml interferon gamma (Gibco, PHC4031). After 4 hr, total RNA was isolated from D21 or T21 cells using TRizol (Invitrogen, 15596026) as specified by the manufacturer and quantified with a Qubit 3.0 using the RNA High Sensitivity (HS) kit (Invitrogen, Q32855). 1 µg of total RNA with an RIN number of ≥8 was used for RNA-seq library prep. Libraries were constructed using Universal Plus mRNA-Seq library preparation kit with NuQuant (Tecan, 0520). Library construction and sequencing were performed at the Genomics Shared Resource (CU Anschutz). Paired-end libraries (151 bp × 151 bp) were sequenced on the Illumina NextSeq 6000 platform (Firmware version 1.26.1, RTA version: v3.4.4, Instrument ID: A00405).

For the non-sibling T21 and sibling T21 *t* = 24 hr RNA-seq experiments, cells were treated with 0.1% DMSO or 100 nM CA for 4.5 hr (non-sibling T21) or 24 hr (sibling T21). After treatment, total RNA was isolated using TRizol (Invitrogen, 15596026) as specified by the manufacturer and quantified with a Qubit 3.0 using the RNA High Sensitivity (HS) kit (Invitrogen, Q32855). 200 ng of total RNA with an RIN number of ≥8 was used for RNA-seq library prep. Libraries were constructed using the Zymo-Seq RiboFree Total RNA Library Kit (Zymo Research). Paired-end sequencing reads of 150 nucleotides were generated on a NovaSeq X (RTA v4.6.7, Serial Number LH00407, Firmware v1.2.0.28691) and de-multiplexed using bcl2fastq.

## RNA-seq computational analysis

For differential gene expression analysis, the workflow was as follows: adaptor sequences were trimmed from the RNA-seq raw fastq files using BBDuk (https://sourceforge.net/projects/bbmap/) and mapped using HISAT2 (*Kim et al., 2019*). Gene counts were generated using featureCounts (*Liao*

*et al., 2014*), and differential expression analysis was performed using DESeq2 (*Anders and Huber, 2010*). When there were multiple transcripts per gene, those with the highest FPKM were kept for analysis. For inter-genotype (T21 vs. D21) comparisons, expression of genes on chromosome 21 was normalized to ploidy. While ERCC RNA Spike-In Mix (Invitrogen, 4456740) was added to isolated RNA samples prior to sequencing, ERCC gene counts were variable across vehicle-treated samples even after accounting for read depth, and when using multiple normalization methods and a linear regression. As such, the median of ratios method native to DESeq2 was used to generate size factors and normalize samples. QIAGEN Ingenuity Pathway Analysis (IPA) version 90348151 (*Krämer et al., 2014*), GSEA 4.2.3 (*Subramanian et al., 2005*), and Gene Ontology analysis (*Ashburner et al., 2000*) were used for identification of activated and inhibited pathways. For IPA analysis, a total of 27,647 out of 28,265 genes could be mapped using Entrez Gene Symbols. For cross-genotype (i.e. T21 vs. D21) pathway analysis, only genes with a fold-change > |1.25| and adjusted p-value <0.1 were used. For within-genotype (i.e. treatment effects) pathway analysis, only genes with an adjusted p-value <0.1 were used.

For analysis of RNA-seq data collected from sibling T21 cell line at 24 hr ±CA, and non-sibling T21 at 4.5 hr ±CA, batch correction was applied to gene counts using Surrogate Variable Analysis (*Leek et al., 2019*) as initial principal component analysis revealed inter-replicate variability as a stronger source of variation than kinase inhibition. However, differential expression analysis demonstrated that the same pathways were predicted to be up- or downregulated regardless of whether batch correction was performed. GSEA was performed by generating a list of genes with differential expression across treatments/genotypes, ranked by the negative log10 adjusted p-value and the sign (positive or negative) of log2(fold-change). Ranked gene lists were loaded onto GSEA (v4.2.3) and GSEA-Preranked was run with the following settings – Number of permutations: 1000, Collapse/Remap: No_collapse, Enrichment statistic: weighted, Max size: 500, Min size: 15.

For splicing analysis, the workflow was as follows: duplicates were removed and adaptors trimmed using the bbTools function (v39.01), trimmed reads were mapped uniquely against the hg38 genome using HISAT2 (v2.1.0) (*Kim et al., 2019*), mapped reads were processed with rMATS (v4.0.1) (*Shen et al., 2014*), and results were filtered based on FDR <0.05, absolute(IncLevelDifference) <0.2, and ≥2 reads/replicate. Sashimi plots were generated from rMATS results using a modified script based on ggsashimi.py.

The IFN score was calculated as described (*Galbraith et al., 2023*) using normalized gene counts from DESeq2. The score is calculated with 18 genes: BPGM, CCL5, IFI27, GMPR, IRF7, CMKLR1, IFITM3, GZMA, FCGR1A, USP18, PLSCR1, CXCL9, IFI44, CD274, CXCL10, ISG15, IFI44L, and RSAD2. For each gene, the mean and standard deviation of the D21 replicates was calculated. Then the measured level of each sample was subtracted by the mean of the D21 replicates and divided by the standard deviation of the D21 replicates to create a standardized gene value. For each sample, the sum of the standardized values was taken to get the IFN score.

The data discussed in this publication have been deposited in NCBI's Gene Expression Omnibus and are accessible through GEO Series accession number GSE220652 or GSE263239. Code used to process and visualize data can be found at https://github.com/kira-alia/Cozzolino2023 (copy archived at *Cozzolino, 2024*).

## Cytokine screen

Cells were pre-treated with 0.1% DMSO or 100 nM CA for 30 min, then treated with 40 mM Tris, pH 7.4 or 10 ng/ml interferon gamma (Gibco, PHC4031). After 24 hr, cells were lysed in RIPA buffer (Thermo Scientific, 89900) supplemented with Halt Protease and Phosphatase Inhibitor Cocktail, EDTA-free (100×) (Thermo Scientific, 78441) and Benzonase endonuclease (Millipore Sigma, 101697). Lysate concentrations were determined using the Pierce BCA Protein Assay Kit (Thermo Scientific, 23225), and 250 µg of protein from each condition was incubated with a membrane array from the Proteome Profiler Human XL Cytokine Array Kit (R&D Systems, ARY022B). Membranes were processed according to the manufacturer's instructions and imaged using an ImageQuant LAS 4000 (GE Healthcare). Background-subtracted technical replicate values for each cytokine in each condition were quantified using ImageJ and averaged giving a normalized intensity value. The relative intensity was compared on a per cytokine basis and was statistically assessed using a one-way ANOVA test between the eight total conditions across two biological replicates.

## PRO-seq

Cells were treated with 0.1% DMSO or 100 nM CA for 75 min before being transferred into a lysis buffer consisting of final concentrations of 10 mM Tris pH 7.4, 2 mM $MgCl_2$, 3 mM $CaCl_2$, 0.5% NP-40, 10% glycerol, 1 mM DTT, 1× Halt Protease and Phosphatase Inhibitor Cocktail (Thermo Scientific, 78441), and 2 U/µl SUPERase•In RNase Inhibitor (Invitrogen, AM2696). Nuclei were isolated and cleaned through repeated centrifugation and washes, and were counted, aliquoted, and snap-frozen in using liquid nitrogen in a freezing buffer consisting of 50 mM Tris pH 8.3, 5 mM $MgCl_2$, 40% glycerol. 0.1 mM EDTA, and 4 U/µl SUPERase•In RNase Inhibitor (Invitrogen, AM2696). Nuclear run-ons were performed for 3 min at 37°C on a Groovin' Tubes thermoshaker (Boekel Scientific, 270500), on a mixture of 10 million human and 100,000 *Drosophila melanogaster* nuclei per replicate. PRO-seq library preparation was performed as described (*Fant et al., 2020*). Library sequencing was performed at the Cornell Genomics Facility (Cornell Institute of Biotechnology) on the Illumina NextSeq 2000 platform (Instrument ID VH00231).

## PRO-seq data analysis

Raw fastq files from the sequencing pipeline described above were trimmed and mapped using the NascentFlow pipeline (DOI 10.17605/OSF.IO/NDHJ2). Bidirectional sites of transcription in the human genome (assembly hg38) were identified using the BidirectionalFlow pipeline (https://github.com/Dowell-Lab/Bidirectional-Flow; *Sanford, 2023*), which runs TFIT (*Azofeifa and Dowell, 2017*). Bidirectional calls across all replicates and conditions relevant to this study were merged using muMerge (https://github.com/Dowell-Lab/mumerge; *Stanley, 2022*) and used as an annotation file for running TFEA (*Rubin et al., 2021*). Example traces of differentially active sites of bidirectional transcription were plotted using pyGenomeTracks (*Lopez-Delisle et al., 2021*).

The data in this publication have been deposited in NCBI's Gene Expression Omnibus and are accessible through GEO Series accession number GSE249549. Code used to process and visualize data can be found at https://github.com/kira-alia/Cozzolino2023 (copy archived at *Cozzolino, 2024*).

# Acknowledgements

We thank A York (Yale University; University of Washington) and P Kovarik (University of Vienna) for helpful comments on the manuscript; we thank Matt Shair (Harvard) for providing Cortistatin A. We acknowledge the Linda Crnic Institute for Down Syndrome Research, particularly the Human Trisome Project Biobank and the Nexus Biobank, for providing the cells used in this work. We thank Theresa Nahreini for cell culture assistance. We acknowledge the Flow Cytometry Shared Core (RRID:S10ODO21601) and other funding support for instrumentation (S10OD025267) at UC-Boulder and the Genomics Shared Resource (RRID:SCR_021984) at UC-Anschutz. This work was supported by the NIH (R01 AI156739 to DJT; R35 GM139550 to DJT; R01 HD100935 to DLB; R01 AI50305 to JME), and the Global Down Syndrome Foundation (JME) and the Anna & John J Sie Foundation (JME, KC, MCSC). KC was supported in part by the NIH (T32 GM065103); KM was supported by the Arnold and Mabel Beckman Foundation and the NSF (MCB-1818147).

# Additional information

### Competing interests

Deepa Ajit: Affiliated to Metabolon Inc, no other competing interest to declare. Joaquín M Espinosa: JME has provided consulting services for Eli Lily and Gilead, and serves on the advisory board of Perha Pharmaceuticals. Robin D Dowell: RDD is a founder of Arpeggio Biosciences. The other authors declare that no competing interests exist.

### Funding

| Funder | Grant reference number | Author |
| --- | --- | --- |
| National Institute of General Medical Sciences | R35GM139550 | Dylan J Taatjes |

| Funder | Grant reference number | Author |
|---|---|---|
| National Institute of Allergy and Infectious Diseases | R01AI156739 | Dylan J Taatjes |
| National Science Foundation | MCB-1818147 | Dylan J Taatjes |
| Eunice Kennedy Shriver National Institute of Child Health and Human Development | R01HD100935 | David Bentley |
| National Institute of Allergy and Infectious Diseases | R01AI50305 | Joaquín M Espinosa |
| National Institute of General Medical Sciences | T32GM065103 | Kira A Cozzolino |
| Arnold and Mabel Beckman Foundation | | Kayla Molison |
| Global Down Syndrome Foundation | | Joaquín M Espinosa |
| Office of the Director | S10OD021601 | Dylan J Taatjes |
| Office of the Director | S10OD025267 | Dylan J Taatjes |

The funders had no role in study design, data collection and interpretation, or the decision to submit the work for publication.

## Author contributions

Kira A Cozzolino, Conceptualization, Data curation, Formal analysis, Investigation, Visualization, Methodology, Writing – review and editing; Lynn Sanford, Resources, Data curation, Formal analysis, Visualization, Methodology; Samuel Hunter, Methodology; Kayla Molison, Formal analysis, Investigation, Methodology; Benjamin Erickson, Data curation, Formal analysis, Investigation, Methodology; Meaghan CS Courvan, Formal analysis, Investigation, Visualization; Taylor Jones, Conceptualization, Data curation, Formal analysis, Investigation, Methodology; Deepa Ajit, Formal analysis, Project administration; Matthew D Galbraith, Data curation, Formal analysis, Methodology; Joaquín M Espinosa, Supervision, Funding acquisition, Project administration; David Bentley, Conceptualization, Supervision, Funding acquisition; Mary Ann Allen, Conceptualization, Methodology; Robin D Dowell, Conceptualization, Formal analysis, Supervision, Funding acquisition, Investigation, Methodology; Dylan J Taatjes, Conceptualization, Formal analysis, Supervision, Funding acquisition, Visualization, Writing – original draft, Project administration, Writing – review and editing

## Author ORCIDs

Matthew D Galbraith ⬦ https://orcid.org/0000-0003-0485-3927
Joaquín M Espinosa ⬦ https://orcid.org/0000-0001-9048-1941
Robin D Dowell ⬦ https://orcid.org/0000-0001-7665-9985
Dylan J Taatjes ⬦ https://orcid.org/0000-0003-4444-5688

Reviewer #1 (Public review): https://doi.org/10.7554/eLife.100197.3.sa1
Reviewer #2 (Public review): https://doi.org/10.7554/eLife.100197.3.sa2
Author response https://doi.org/10.7554/eLife.100197.3.sa3

# Additional files

## Supplementary files
MDAR checklist

## Data availability
The data in this publication have been deposited in NCBI's Gene Expression Omnibus and are accessible through GEO Series accession number GSE249549, GSE220652, or GSE263239.

The following datasets were generated:

| Author(s) | Year | Dataset title | Dataset URL | Database and Identifier |
|---|---|---|---|---|
| Cozzolino KA | 2024 | Mediator kinase inhibition suppresses hyperactive interferon signaling in Down syndrome II | https://www.ncbi.nlm.nih.gov/geo/query/acc.cgi?acc=GSE249549 | NCBI Gene Expression Omnibus, GSE249549 |
| Cozzolino KA, Erickson B, Bentley D | 2023 | Mediator kinase inhibition suppresses hyperactive interferon signaling in Down syndrome | https://www.ncbi.nlm.nih.gov/geo/query/acc.cgi?acc=GSE220652 | NCBI Gene Expression Omnibus, GSE220652 |
| Cozzolino KA, Courvan MCS | 2025 | Mediator kinase inhibition suppresses hyperactive interferon signaling in Down syndrome III | https://www.ncbi.nlm.nih.gov/geo/query/acc.cgi?acc=GSE263239 | NCBI Gene Expression Omnibus, GSE263239 |

The following previously published dataset was used:

| Author(s) | Year | Dataset title | Dataset URL | Database and Identifier |
|---|---|---|---|---|
| Galbraith MD, Rachubinski AL, Smith KP, Sullivan KD, Espinosa JM | 2023 | Crnic Institute Human Trisome Project: PolyA RNA-sequence from whole blood | https://www.ncbi.nlm.nih.gov/geo/query/acc.cgi?acc=GSE190125 | NCBI Gene Expression Omnibus, GSE190125 |

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
