## [Editor Report · eLife Assessment]

This is an **important** study providing **compelling** evidence that the Mediator kinase module mediates an elevated inflammatory response, manifested by heightened cytokine levels, associated with Downs syndrome (DS) via transcriptional changes impacting cell signaling and metabolism that involve mobilization of nuclear receptors by altered lipid metabolites, which has significance for the treatment of DS and other chronic inflammatory conditions. Particular strengths of the study include the combined experimental approaches of transcriptomics, untargeted metabolomics and cytokine screens and the use of sibling matched cell lines (trisomy 21 vs disomy 21) from various donors. Evidence is also provided implicating the Mediator kinase module in controlling mRNA splicing and mitochondrial function that should stimulate new research to elucidate the mechanistic bases for these novel functions.

---

## [Referee Report · Reviewer #1 (Public review)]

Summary:

The main conclusion of this manuscript, that the mediator kinases supporting the IFN response in Downs syndrome cell lines, represents an important addition to understanding the pathology of this affliction.

Strengths:

Mediator kinase stimulates cytokine production. Both RNAseq and metabolomics clearly demonstrate a stimulatory role for CDK8/CDK19 in the IFN response. The nature of this role, direct vs. indirect, is inferred by previous studies demonstrating that inflammatory transcription factors are Cdk8/19 substrates. The cytokine and metabolic changes are clear cut and provide a potential avenue to mitigate these associated pathologies.

Weaknesses:

Seahorse analysis is normally calculated with specific units for oxygen consumption, ATP production, etc. It would be of interest to see the actual values of OCR (e.g., pmol/O2 consumption/number of cells) between the D21 and T21 cell lines rather than standardizing the results. Previous studies reported reduced mitochondrial function with DS cell lines and model systems e.g., see [10.1016/j.bbadis.2022.166388] and aberrant mitochondrial morphology/oxidative stress [10.1016/j.cmet.2012.12.005] [10.1016/j.neuroscience.2022.12.003]. This report observes elevated mitochondrial function in the T21 cells vs. the D21 control. There are several potential reasons for these differences but it is not up to the authors to rectify their results with others. However, it would be of interest to the general reader that they be mentioned in the discussion.

---

## [Referee Report · Reviewer #2 (Public review)]

Summary:

In this manuscript, Cozzolino et al. demonstrate that inhibition of the Mediator kinase CDK8 and its paralog CDK19 suppresses hyperactive interferon (IFN) signaling in Down syndrome (DS), which results from trisomy of chromosome 21 (T21). Numerous pathologies associated with DS are considered direct consequences of chronic IFN pathway activation, and thus hyperactive IFN signaling lies at the heart of pathophysiology. The collective interrogation of transcriptomics, metabolomics, and cytokine screens in sibling-matched cell lines (T21 vs D21) allows the authors to conclude that Mediator kinase inhibition could mitigate chronic, hyperactive IFN signaling in T21. To probe the functional outcomes of Mediator kinase inhibition, the authors performed cytokine screens, transcriptomic, and untargeted metabolomics. This collective approach revealed that Mediator kinases establish IFN-dependent cytokine responses at least in part through transcriptional regulation of cytokine genes and receptors. Mediator kinase inhibition suppresses cell responses during hyperactive IFN signaling through inhibition of pro-inflammatory transcription factor activity (anti-inflammatory effect) and alteration of core metabolic pathways, including upregulation of anti-inflammatory lipid mediators, which served as ligands for specific nuclear receptors and downstream phenotypic outcomes (e.g., oxygen consumption). These data provided a mechanistic link between Mediator kinase activity and nuclear receptor function. Finally, the authors also disclosed that Mediator kinase inhibition alters splicing outcomes.

Overall, this study reveals a mechanism by which Mediator kinases regulate gene expression and establish that its inhibition antagonizes chronic IFN signaling through collective transcriptional, metabolic, and cytokine responses. The data have implications for DS and other chronic inflammatory conditions, as Mediator kinase inhibition could potentially mitigate pathological immune system hyperactivation.

Comments on revisions:

In the record of version, the authors have improved readability and also incorporated experiments that provide compelling support to the main discovery of the story. Below I summarize the previous strengths and how they improved noted weaknesses.

(1) One major strength of this study is the mechanistic evidence linking Mediator kinases to hyperactive IFN signaling through transcriptional changes impacting cell signaling and metabolism.

(2) Another major strength of this study is the use of sibling matched cell lines (T21 vs D21) from various donors (not just one sibling pair), and further cross-referencing with data from large cohorts, suggesting that part of the data and conclusions are generalizable.

(3) Another major strength of this study is the combined experimental approach including transcriptomics, untargeted metabolomics and cytokine screens to define the mechanisms underlying suppression of hyperactive interferon signaling in DS upon Mediator kinase inhibition.

(4) Another major strength of this study is the significance of the work to DS and its potential impact to other chronic inflammatory conditions.

(5) The previously noted weakness regarding the roles of nuclear receptors to activation of an anti-inflammatory program upon Mediator kinase inhibition was not directly experimentally addressed because existing data from other studies (referenced in this version) have linked specific nuclear receptors to lipid biosynthesis and anti-inflammatory cascades. This is considered acceptable.

(6) The presentation of the splicing data analysis is not better integrated in the overall story.

(7) The authors improved the readability of the manuscript by providing specific details throughout.

(8) Figures were improved and simplified when possible to facilitate readability.

(9) The authors now clarified the PRO-Seq (TFEA analysis) explaining that their data is consistent with the general observation that stimulus-responsive genes is controlled by enhancer-bound TFs.

---

## [Author Response]

The following is the authors’ response to the original reviews.

**Reviewer #1 (Public Review):**
Summary:The main conclusion of this manuscript is that the mediator kinases supporting the IFN response in Downs syndrome cell lines represent an important addition to understanding the pathology of this affliction.Strengths:Mediator kinase stimulates cytokine production. Both RNAseq and metabolomics clearly demonstrate a stimulatory role for CDK8/CDK19 in the IFN response. The nature of this role, direct vs. indirect, is inferred by previous studies demonstrating that inflammatory transcription factors are Cdk8/19 substrates. The cytokine and metabolic changes are clear-cut and provide a potential avenue to mitigate these associated pathologies.Weaknesses:This study revealed a previously undescribed role for the CKM in splicing. The previous identification of splicing factors as substrates of CDK8/CDK19 is also intriguing. However, additional studies seem to be necessary in order to attach this new function to the CKM. As the authors point out, the changes in splicing patterns are relatively modest compared to other regulators. In addition, some indication that the proteins encoded by these genes exhibit reduced levels or activities would support their RNAseq findings.

We have added new splicing data for the version of record. Specifically, we have added splicing data analysis for the "non-sibling" T21 cell line (±cortistatin A, t=4.5h) and for the sibling T21 line (±cortistatin A) at t=24h. The results are summarized in new Figure 5 – figure supplement 2. The data are in agreement with our prior data from the sibling T21 line ±CA at t=4.5h. In particular, (i) similar numbers of genes were impacted by splicing changes (alternative exon inclusion or alternative exon skipping) in CA-treated cells in the "non-sibling" T21 line compared with the sibling T21 line; (ii) upon completion of a pathway analysis of these alternatively spliced genes, similar pathways were affected by CA in each case (non-sibling T21 vs. sibling T21), in particular those related to IFN signaling; (iii) regarding the new t=24h timepoint for the sibling T21 line, similar numbers of genes were alternatively spliced (alternative exon inclusion or alternative exon skipping) in CA-treated cells compared with the 4.5h timepoint, and (iv) the IPA results with the alternately spliced genes identified inflammatory signaling, mRNA processing, and lipid metabolism among other pathways, which broadly reflect the cytokine screen and metabolomics data in CA-treated cells (t=24h).

Additional evidence for CDK8/CDK19 regulation of splicing comes from our t=24h RNA-seq data in T21 cells ±CA. GSEA results revealed down-regulation of many pathways related to RNA processing and splicing, suggesting that the splicing changes caused by Mediator kinase inhibition result from reduced expression of splicing regulators, at least at this longer timeframe. These results are summarized in new Figure 2 – figure supplement 2E. Collectively, the data shown in this article reveal a previously unidentified role for Mediator kinases as splicing regulators. We emphasize in the article, however, that the splicing effects of Mediator kinase inhibition appear modest, at least within the cell lines and timeframes of our experiments, especially when compared with CDK7 inhibition [Rimel et al. Genes Dev 2020 1452].

Seahorse analysis is normally calculated with specific units for oxygen consumption, ATP production, etc. It would be of interest to see the actual values of OCR between the D21 and T21 cell lines rather than standardizing the results. This will address the specific question about relative mitochondrial function between these cells. Reduced mitochondrial function has been associated with DS patients. Therefore, it would be important to know whether mitochondrial function is reduced in the T21 cells vs. the D21 control. Importantly for the authors' goal of investigating the use of CDK8/19 inhibitors in DS patients, does CA treatment reduce mitochondrial function to pathological levels?

These are good points. We have addressed as follows.

(1) We have added a comparative analysis of Seahorse data for the sibling-matched T21 and D21 lines. As shown in new Figure 2 – figure supplement 4A-C, the T21 line shows higher basal levels of OCR and ECAR compared with D21. Although reviewer 1 states that "reduced mitochondrial function has been associated with DS patients" we are unaware of the study from which this conclusion was made. Our results are consistent with a Down syndrome mouse model study published last year [Sarver et al. eLife 2023 e86023]. We acknowledge that in this study, T21/D21 OCR levels varied in different tissues, but the majority of tissue types showed elevated OCR in T21, similar to our results in the human B-cells used here.

(2) Interestingly, CA treatment reduced OCR and ECAR in T21 cells (and D21), suggesting that Mediator kinase inhibition might normalize mitochondrial function (and ECAR) toward D21 levels. We show this comparison in new Figure 2 – figure supplement 4D-F. Indeed, CA treatment appears to normalize T21 mitochondrial function and ECAR toward D21 levels. Although this may suggest a therapeutic benefit, we emphasize that more experiments would be needed to make such claims with confidence.

(3) We include a breakdown of mitochondrial parameters from Seahorse data in the bar plots shown in Figure 2–figure supplement 3. This includes ATP production, which shows reduced ATP levels in CA-treated T21 cells specifically.

(4) We have added Seahorse data for ECAR (extracellular acidification rate) in the siblingmatched D21 and T21 cells, ±CA. These results are shown in new Figure 2 – figure supplement 3D, and indicate that CA treatment reduces ECAR in both D21 and T21 cells. This result is consistent with a prior report that analyzed ECAR in CDK8 analog-sensitive HCT116 cells [Galbraith et al. Cell Rep 2017 1495].

**Reviewer #2 (Public Review):**
Summary:In this manuscript, Cozzolino et al. demonstrate that inhibition of the Mediator kinase CDK8 and its paralog CDK19 suppresses hyperactive interferon (IFN) signaling in Down syndrome (DS), which results from trisomy of chromosome 21 (T21). Numerous pathologies associated with DS are considered direct consequences of chronic IFN pathway activation, and thus hyperactive IFN signaling lies at the heart of pathophysiology. The collective interrogation of transcriptomics, metabolomics, and cytokine screens in sibling-matched cell lines (T21 vs D21) allows the authors to conclude that Mediator kinase inhibition could mitigate chronic, hyperactive IFN signaling in T21. To probe the functional outcomes of Mediator kinase inhibition, the authors performed cytokine screens, transcriptomic, and untargeted metabolomics. This collective approach revealed that Mediator kinases establish IFN-dependent cytokine responses at least in part through transcriptional regulation of cytokine genes and receptors. Mediator kinase inhibition suppresses cell responses during hyperactive IFN signaling through inhibition of pro- inflammatory transcription factor activity (anti-inflammatory effect) and alteration of core metabolic pathways, including upregulation of anti-inflammatory lipid mediators, which served as ligands for specific nuclear receptors and downstream phenotypic outcomes (e.g., oxygen consumption). These data provided a mechanistic link between Mediator kinase activity and nuclear receptor function. Finally, the authors also disclosed that Mediator kinase inhibition alters splicing outcomes.Overall, this study reveals a mechanism by which Mediator kinases regulate gene expression and establish that its inhibition antagonizes chronic IFN signaling through collective transcriptional, metabolic, and cytokine responses. The data have implications for DS and other chronic inflammatory conditions, as Mediator kinase inhibition could potentially mitigate pathological immune system hyperactivation.Strengths:(1) One major strength of this study is the mechanistic evidence linking Mediator kinases to hyperactive IFN signaling through transcriptional changes impacting cell signaling and metabolism. (2) Another major strength of this study is the use of sibling-matched cell lines (T21 vs D21) from various donors (not just one sibling pair), and further cross-referencing with data from large cohorts, suggesting that part of the data and conclusions are generalizable.(3) Another major strength of this study is the combined experimental approach including transcriptomics, untargeted metabolomics, and cytokine screens to define the mechanisms underlying suppression of hyperactive interferon signaling in DS upon Mediator kinase inhibition. (4) Another major strength of this study is the significance of the work to DS and its potential impact on other chronic inflammatory conditions.Weakness:(1) Genetic evidence linking the mentioned nuclear receptors to activation of an anti-inflammatory program upon Mediator kinase inhibition could improve the definition of the mechanism and overall impact of the work.

Existing data from other studies, some of which are cited in the article, have linked PPAR and LXR to lipid biosynthesis and anti-inflammatory signaling cascades. We assume that reviewer 2 is suggesting knockdown and/or degron depletion of specific nuclear receptors, to compare/contrast the effect of CA on IFN responses in T21 and D21 cells. Such experiments would help de-couple the NR-specific contributions from other CA-dependent effects. We consider these experiments important next steps for this project, but beyond the scope of this study. That said, we anticipate that data from such experiments might be challenging to interpret, given the complex and inter-connected cascade of transcriptional and metabolic changes that would result from PPAR or LXR depletion.

(2) Page 5 states that "Mediator kinases broadly regulate cholesterol and fatty acid biosynthesis and this was further confirmed by the metabolomics data", but a clear mechanistic explanation was lacking. Likewise, the data suggest but do not prove, that altered lipid metabolites influence the function of nuclear receptors to regulate an anti-inflammatory program in response to Mediator kinase inhibition (p. 6), despite the fact the gene expression changes elicited by Mediator kinase inhibition tracked with downstream metabolic changes.

We have clarified the text on page 5 to address this comment. Specifically, we note that CA treatment increases expression of FA metabolism and cholesterol metabolism genes in T21 cells under basal conditions, and the genes affected are shown in Figure 2–figure supplement 1E. Thus, the mechanistic explanation is that Mediator kinases cause elevated levels of FA and cholesterol metabolites via changes in expression of FA and cholesterol biosynthesis genes (at least in part). We further address the mechanism with the PRO-seq data and TFEA results in Figure 6; in particular, p53 activity is rapidly suppressed in CA-treated T21 cells (t=75min), and this alone is sufficient to activate SREBP [Moon et al. Cell 2019 564]. CA-dependent activation of SREBP target genes is a dominant feature in the T21 RNA-seq data (t=4.5h).

We agree with the second point raised by reviewer 2, that our data suggest but do not prove nuclear receptor function is altered by CA treatment. We do cite papers that have provided good evidence that the metabolites elevated in CA-treated cells are NR ligands and activate their target genes. Additional experiments to address this question might involve targeted depletion of select metabolites via inhibition of key biosynthetic enzymes. We consider these experiments beyond the scope of this already expansive article. That said, it will be challenging to conclusively demonstrate clear cause-effect relationships (e.g. to demonstrate whether select metabolites altered by CA treatment directly alter PPARA function), given (i) the myriad transcriptional and metabolic changes caused by CA treatment, coupled with the fact that (ii) the CA-dependent lipid metabolite changes are spread out across chemically distinct NR agonists (e.g. endocannabinoids, oleamide, or cholesterol metabolites such as desmosterol), and (iii) NR activation can occur via multiple different metabolites.

(3) The figures are outstanding but dense.

Thank you. We have done our best to represent the results clearly and within the publication guidelines. There was an enormous amount of data to summarize for this article.

(4) Figure 6 (PRO-Seq). The authors refer to pro-inflammatory TFs (e.g. NF-kB/RelA). It is not clear whether the authors have specifically examined TF binding at enhancers or more broadly at every region occupied by the interrogated TFs?

This is a good point. Our analysis (TFEA) only identified the TFs whose activity was changing in CA-treated cells. It did not distinguish where these TFs were bound (enhancers vs. promoters). We completed a modified TFEA by separating enhancer TFs vs. promoter TFs. The results showed a preference for CA-dependent suppression of enhancer-bound TFs. This result is consistent with the general observation that stimulus-response transcription is controlled by enhancer-bound TFs (e.g. Kim et al. Nature 2010 182; Azofeifa et al. Genome Res 2018 334; Jones et al. bioRxiv 2024 585303). However, our TFEA enhancer/promoter analysis is preliminary and more work would be needed to address this comment in a rigorous way. Therefore, we did not include this analysis in the revision.

**Reviewing Editor Comments:**
Main suggestions for improvement:(1) Provide additional information about the mechanistic basis for the changes in lipid levels observed on kinase inhibition.

We have changed the text to better emphasize that the mechanistic basis involves (i) gene expression changes resulting from Mediator kinase inhibition (e.g. Fig 2 – figure supplement 1D, E, Fig 2 – figure supplement 2B, Fig 2 – figure supplement 4B-D); (ii) activation of SREBP and PPAR and LXR, based upon IPA results with RNA-seq data (e.g. Fig 2B, Fig 2 – figure supplement 1F, Fig 2 – figure supplement 2D, Fig 2 – figure supplement 4E; Fig 3E), and (iii) rapid CAdependent suppression of p53 function (Fig 6A), which will activate SREBP (Moon et al. Cell 2019 564).

(2) Provide direct genetic evidence that the nuclear receptors are activated by the lipid changes to mediate an anti-inflammatory program in response to Mediator kinase inhibition.

This is an excellent question but we consider it beyond the scope of this already expansive study. That said, we cite several papers in the article that demonstrate that the lipids we observe elevated in CA-treated cells (i) directly bind PPAR or LXR and (ii) activate their TF function. We also note that the anti-inflammatory impacts of Mediator kinase inhibition are broad, affecting distinct gene sets through transcriptional changes, metabolites, and cytokines. Any NR-specific contributions could be challenging to de-couple from CA-dependent effects using knockdown or depletion methods, given the compensatory responses that would result.

(3) Improve/expand the evidence that Mediator kinase inhibition confers reduced mitochondrial function.

We have added new Seahorse data for sibling-matched D21 and T21 cells (±CA) for the version of record. Our prior results showed reduced mitochondrial function and OCR in CA-treated T21 cells. We have added data that compares D21 and T21 mitochondrial function. As shown in new Figure 2 – figure supplement 4A-C, the T21 line shows higher basal levels of OCR and ECAR compared with D21. These results are consistent with a Down syndrome mouse model study published last year [Sarver et al. eLife 2023 e86023]. When we compare CA-treated T21 with D21 cells, mitochondrial respiration and OCR are similar, suggesting that Mediator kinase inhibition might normalize mitochondrial function (and ECAR) toward D21 levels. We show this comparison in new Figure 2 – figure supplement 4D-F. Although this may suggest a therapeutic benefit, we emphasize that more experiments would be needed to make such claims with confidence.

(4) Determine whether mitochondrial function is reduced in the T21 cells vs. the D21 controls and whether kinase inhibition with the inhibitor reduces mitochondrial function to pathological levels.

For the version of record, we have added a direct comparison of mitochondrial parameters and OCR in the sibling-matched D21/T21 lines. The data show that T21 cells have higher OCR compared with D21. These results are consistent with a Down syndrome mouse model study published last year [Sarver et al. eLife 2023 e86023]. Our results also indicate that CA treatment brings OCR and other "mitochondrial parameters" in T21 cells toward D21 levels, as noted above.

(5) Consider whether the CDK8/19 inhibitor has off-target effects that would lessen its therapeutic value.

We chose cortistatin A (CA) for this project because it is the most potent and selective inhibitor available for targeting CDK8/CDK19. Initial published reports suggested off-target effects (Cee et al. Angew Chem IEE 2009), but these experiments used binding assays against the kinase protein alone, and did not measure binding or inhibition with biologically relevant, active kinase complexes. Kinome-wide screens involving native, active kinase complexes showed no evidence of off-target effects for cortistatin A, even at concentrations 5000-times the measured KD (Pelish et al. Nature 2015). See Author response image 1.

Related to CA therapeutic value, that is an important issue but beyond the scope of this study. We consider CA a valuable chemical probe, to use as a means to define CDK8/CDK19-dependent functions in cell line models. As a chemical probe, we consider CA the "best-in-class" Mediator kinase inhibitor, based upon all available data (Clopper & Taatjes Curr Opin Chem Biol 2022 102186).

That said, we understand the concern about off-target effects, which can never be ruled out with a chemical inhibitor. We include quantitative western data (Fig 1 – figure supplement 1A) that compares CA with a structurally distinct CDK8/CDK19 inhibitor, CCT251545. The data show that, as expected, CA (100nM) and CCT251545 (250nM) similarly inhibit STAT1 S727 phosphorylation in IFN-stimulated cells. The samples were pre-treated with inhibitor for 30 minutes prior to IFNg and collected 45 minutes after IFNg treatment.

We did not complete any experiments with knockouts or kinasedead alleles primarily because knockouts or kinase-dead alleles are not reliable comparisons for chemical inhibition because of the different time frames involved. For example, there will be genetic compensation in edited cell lines (Rossi/Stanier Nature 2015 230) and we and others have shown that there are major differences between kinase protein loss through knockdown or knockout methods vs. rapid inhibition with small molecules (e.g. Poss et al. Cell Rep 2016 436; Sooraj et al. Mol Cell 2022 123).

**Author response image 1. sa3fig1:** Information about cortistatin A. (A) KiNativ kinome screen from HEK293 lysates. CA blocked capture of only CDK8/CDK19 in this MSbased assay, among over 200 kinases detected. (B) Equilibrium binding constants and kinetics for CA. (C) CA structure; note the dimethylamine is protonated at physiological pH, and forms a pi-cation interaction with W105 (crystal structure, panel D). Only CDK8 and CDK19 have an aromatic residue (W) at this position, providing a structural basis for high selectivity.

(6) Improve the presentation of the splicing data and better discuss how the splicing alterations may be contributing to the disease phenotype.

We have added new splicing data for the version of record. Specifically, we have added splicing data analysis for the "non-sibling" T21 cell line (±cortistatin A, t=4.5h) and for the sibling T21 line (±cortistatin A) at t=24h. The results are summarized in new Figure 5 – figure supplement 2. The data are in agreement with our prior results from the sibling T21 line ±CA at t=4.5h. In particular, (i) similar numbers of genes were impacted by splicing changes (alternative exon inclusion or alternative exon skipping) in CA-treated cells in the "non-sibling" T21 line compared with the sibling T21 line; (ii) upon completion of a pathway analysis of these alternatively spliced genes, similar pathways, including IFN signaling pathways, were affected by CA in each case (non-sibling T21 vs. sibling T21); (iii) regarding the new t=24h timepoint for the sibling T21 line, similar numbers of genes were alternatively spliced (alternative exon inclusion or alternative exon skipping) in CA-treated cells compared with the 4.5h timepoint, and (iv) the IPA results with the alternately spliced genes identified inflammatory signaling, mRNA processing, nucleotide and lipid metabolism among other pathways, which broadly reflect the cytokine screen and metabolomics data in CA-treated cells (t=24h).

Additional evidence for CDK8/CDK19 regulation of splicing comes from our t=24h RNA-seq data in T21 cells ±CA. GSEA results from sibling T21 cells ±CA revealed down-regulation of many pathways related to RNA processing and splicing (RNA-seq data, t=24h), suggesting that the splicing changes caused by Mediator kinase inhibition result from reduced expression of splicing regulators, at least at longer timeframes. These results are summarized in new Figure 2 – figure supplement 2E.

Related to how splicing alterations may be contributing to the CA-dependent effects and their potential therapeutic implications, this is an interesting question but open-ended. It will not be straightforward to link specific splicing changes to possible therapeutic outcomes, especially given that there are hundreds of genes affected and because the effects are modest (i.e. not all-ornothing).

**Reviewer #1 (Recommendations For The Authors):**
The findings that CA treatment leads to upregulation of as many genes are downregulated is consistent with previous studies of a 50:50 role for the CKM. However, most previous studies utilized knockout alleles or knockdown approaches. As the authors demonstrated in a previous study, CA inhibits kinase activity without changing CDK8 levels. Does this indicate that the kinase activity of Cdk8/19 is required for transcriptional repression? Previous in vitro studies suggested that Cdk8/19-dependent repression was independent of their kinase activity. The authors should comment on this.

This is a challenging question to address, because the answer will depend on the timing of the experiment and the experimental context. The short answer is that the kinase activity of CDK8/19 will activate some genes and reduce expression of others, at least in part because CDK8/19 phosphorylate TFs, which drive global gene expression programs. TF phosphorylation by CDK8/19 appears to activate some genes and repress others (e.g. STAT1 S727A example from Steinparzer et al. Mol Cell 2019 485), at least based upon RNA-seq data, but this doesn't measure the immediate effects on the transcriptome. It is true that kinase activity isn't required to block pol II incorporation into the PIC (Knuesel et al. Genes Dev 2009 439). This is a kinase-independent function of the module; MKM-Mediator binding will block Mediator-pol II interaction and therefore block PIC assembly and pol II initiation (Knuesel 2009; Ebmeier & Taatjes PNAS 2010 11283). The kinase-independent functions of CDK8/19 were not a focus of the work described here. We only focus on Mediator kinase activity. We also do not focus on potential effects on RNAPII initiation or PIC assembly, although these are important peripheral topics.

Descriptors are less useful as the reader must go back to reconstruct the experiment: "Although metabolites were measured 24h after CA treatment, these data suggest that altered lipid metabolites influence LXR and PPAR function". Does "altered" mean the lipid concentrations were up or down? Similarly, lipids that "influenced" LXR function - were they stimulatory or inhibitory?

Good point. Where possible, we used more accurate language when describing CAdependent changes.

I found many sections in the text confusing. For example: Figure 3. Mediator kinase inhibition antagonizes IFNγ transcriptional responses in T21 and D21. It takes a while to unpack this figure title. Instead of the double negative, the authors could simply state that "Mediator kinase is required for IFN-dependent transcriptional activation". Describing the protein activity, versus the drug-induced phenotype, can often clarify complicated scenarios.

Good idea. We have edited the text to eliminate some but not all of these double negatives. In some cases we prefer to describe the consequence of kinase inhibition.

**Reviewer #2 (Recommendations For The Authors):**
(1) The splicing data analysis is compelling, but not well integrated into the overall story and it cuts the storytelling logic in the Abstract. The authors could consider better integrating the large amount of data generated and better explaining how it relates to the various aspects of the proposed model (transcriptional, metabolism) to help improve potential cause-and-effect outcomes. -

We agree. The large amount of data, combined with the different experimental approaches, makes it a challenge to summarize the data in a concise way. We have done our best to organize the results in a logical and clear manner. To address this comment, we have gone through the text and re-organized where possible, and we have edited the abstract. We have added new splicing data and the splicing results are now better integrated (in our opinion) in part because of the pathway results from the t=24h ±CA RNA-seq data, which show major reductions in gene sets related to splicing and RNA processing.

(2) The manuscript could improve its readability by providing specific details throughout. Examples include (i) explaining why and what 29 cytokines were chosen for the screen (p. 3, p. 4) (ii) providing major data analysis conclusions to the cytokine screen part (p. 3) (iii) expanding the conclusions to the metabolic pathway analysis (p. 4) (iv) being more precise when referring to T21-specific changes (up or down?) (p 4), and "significantly altered" by CA treatment in T21 cells (up or down?) (p. 5).

Good points. We have edited the text to address these comments. Please note that the 29 cytokines refers to a different study (Malle et al. Nature 2023) and we had no role in selecting the cytokines. Our screen involved 105 cytokines that were arrayed as part of a commercially available panel.

(3) The figures are outstanding but dense (e.g., Figure 1b, can any simplification and/or highlighting be done to underscore important features?). Some panels are illegible (e.g. Figure 1- supplement Figure 2a and b). The authors could improve data presentation. For example, the Venn diagrams (e.g., Figure 2f) are hard to quickly digest. Can the authors find a better way to highlight important data (e.g., hard to distinguish the meaning of font bolding from italics)?

Thank you for these suggestions. Regarding Figure 1B, we simplified the metabolic pathways to emphasize the biochemicals that specifically relate to this study. We decided against highlighting specific metabolites beyond this simplification, because in our opinion it causes as many problems as it solves. Where possible, we have enlarged the panels with hard-to-read text; thank you for the suggestion. For the Venn diagrams, they convey a large amount of information in a single panel: increased or decreased gene expression in T21 or D21, cytokine genes or cytokine receptors, and gene expression convergence or divergence compared with protein levels from cytokine screens. There is a different way to display the results, but it would involve generating more data panels to parse out the results. This could be considered better, but we opted for something that is more information-rich that requires only a single data panel. Given the large amount of data already shown, we hope the reviewer can understand this choice.